# ViewPoint: Panoramic Video Generation with Pretrained Diffusion Models

**Zixun Fang**[1,2*]  **Kai Zhu**[1,2]  **Zhiheng Liu**[3]  **Yu Liu**[2]  **Wei Zhai**[1]
**Yang Cao**[1]  **Zheng-Jun Zha**[1†]

[1] MoE Key Laboratory of Brain-inspired Intelligent Perception and Cognition,
University of Science and Technology of China
[2] TongYi Lab    [3] HKU

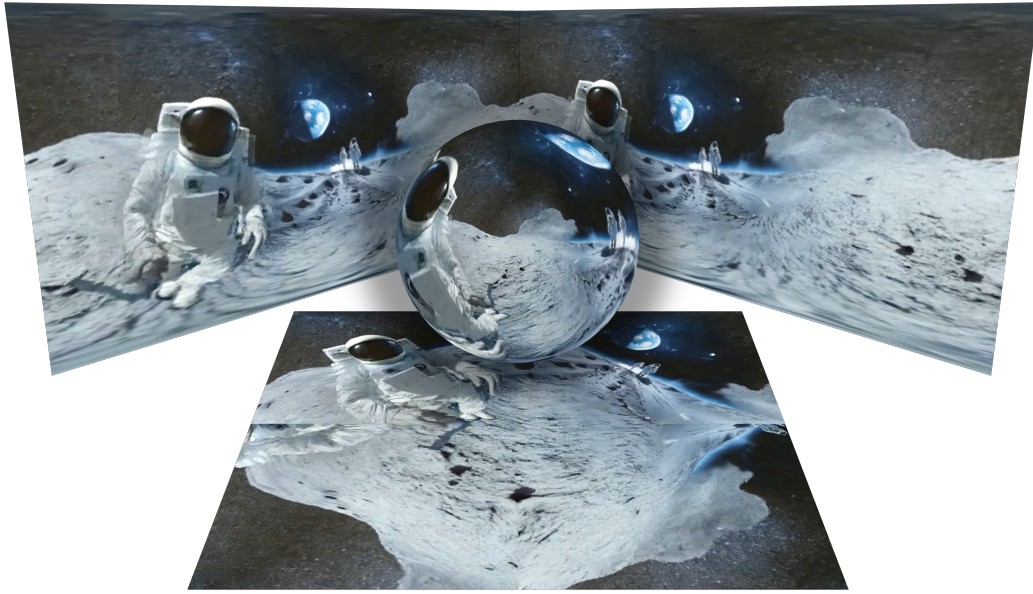

Figure 1: **The generated result.** The image at the bottom displays the ViewPoint map generated by our model, with the background image showing the concatenated equirectangular map derived from the ViewPoint map. The generated panoramic image exhibits excellent spatial consistency, as the equirectangular map can be seamlessly stitched together. Project page: ViewPoint.

## Abstract

Panoramic video generation aims to synthesize 360-degree immersive videos, holding significant importance in the fields of VR, world models, and spatial intelligence. Existing works fail to synthesize high-quality panoramic videos due to the inherent modality gap between panoramic data and perspective data, which constitutes the majority of the training data for modern diffusion models. In this paper, we propose a novel framework utilizing pretrained perspective video models for generating panoramic videos. Specifically, we design a novel panorama representation named ViewPoint map, which possesses global spatial continuity and fine-grained visual details simultaneously. With our proposed Pano-Perspective attention mechanism, the model benefits from pretrained perspective priors and captures the panoramic spatial correlations of the ViewPoint map effectively. Extensive experiments demonstrate that our method can synthesize highly dynamic

---

*Work done at TongYi Lab.
†Corresponding Author

and spatially consistent panoramic videos, achieving state-of-the-art performance and surpassing previous methods.

# 1 Introduction

Imagine you're traveling—would you seize the chance to capture the breathtaking landscapes that unfold before you? When you return home, do you long to relive and immerse yourself in those vivid experiences once more? Recently, omnidirectional vision has garnered increasing attention as it unlocks immersive AR/VR, virtual travel, and telepresence experiences. However, recording 360-degree videos requires expensive professional devices, *i.e.,* 360 cameras, making the creation of panoramic content challenging, and most consumers capture only narrow field-of-view (FoV) clips on portable monocular cameras, *e.g.,* smartphones. Enabling these perspective recordings to become full panoramas would democratize spherical media, letting anyone relive or share memories in true 360-degree form.

Achieving this conversion is non-trivial due to fundamental representation gaps between perspective and panorama domains. A straightforward approach is to adopt the widely-used Equirectangular Projection (ERP) for panoramas, which maps the spherical view onto a rectangular image. Unfortunately, this format introduces severe distortions, especially near the poles, stretching and squashing content unnaturally. More critically, equirectangular images lie outside the distribution of typical training data for modern generative models, which are at the center of the recent surge of interest in computer vision and pattern recognition [45, 27, 28, 42, 41, 19–24]. Diffusion models [10, 29, 32] and VAEs [14] are predominantly trained on perspective imagery, so without significant adaptation they struggle to produce high-quality results in the warped ERP space. On the other hand, one could represent the panorama as multiple perspective projections. A common choice is the Cubemap Projection(CP) format, which unfolds the sphere into six faces (each a 90° FOV perspective view). The CP representation avoids polar distortions and yields locally planar patches well-aligned with the priors of convolutional and diffusion networks. However, a naive cubemap suffers from spatial discontinuities at the borders of the faces. The six faces are disjoint on a 2D grid, making it difficult for a neural model to capture cross-face consistency. In short, existing representations force a trade-off between continuity and distortion: ERP offers end-to-end continuity but distorts the content, whereas CP preserves local fidelity but fragments the panoramic space. Neither is ideal for generative video modeling, especially when temporal consistency is also required.

In this paper, we address the above-mentioned issues by introducing ViewPoint, a novel 360° video representation and generation framework that bridges the strengths of ERP and CP. At the core of ViewPoint is a spatially-aware pseudo-stitching scheme that reprojects and rearranges the scene into an overlapping set of perspective views with greatly improved continuity. Specifically, we first convert the panorama into six cube faces—each a distortion-free perspective view—and then merge them into a small number of overlapping "pseudo-perspective" panels. Because these panels share content along their boundaries, they eliminate the hard seams of a standard cubemap while remaining fully compatible with existing 2D diffusion models. On this representation, the proposed Pano-Perspective attention delivers two key benefits: (1) Global coherence: Pano-attention blocks span the entire stitched map (and time), ensuring that opposite directions align and the scene stays consistent. (2) Local fidelity: Perspective-attention blocks focus on each panel's neighborhood, preserving fine texture, color, and motion details. With this attention mechanism, we take full advantage of the generative capabilities of diffusion models and adapt them to the task of panoramic video generation, where high-quality data is scarce.

In summary, our contributions are as follows:

- We introduce a novel representation for 360-degree content aimed at improving spatial continuity and reducing distortion while leveraging the power of diffusion models to generate 360-degree videos through the proposed format.

- We design a Pano-Perspective attention mechanism, enabling the model to simultaneously maintain global spatial continuity across the entire panorama and significantly enhance the preservation of fine-grained details and motion.

- Extensive experiments demonstrate that our method can generate high-quality 360-degree videos and achieve state-of-the-art performance, outperforming previous approaches.

## 2 Related Works

**Panorama Representations.** Panoramic images are often projected from spherical space to the 2D plane for processing and storage [1]. **Equirectangular Projection (ERP)** is the most popular format, which uniformly maps pixels from a spherical surface to a planar rectangle. Despite its simplicity and spatial continuity, ERP inevitably brings disadvantages such as geometric distortion at the poles. **Cubemap Projection (CP)**, on the other hand, projects panoramas to six cube faces with the FoV of $90° \times 90°$ to alleviate geometric distortion. However, under this representation, only one cube face is explicitly spatially continuous with its four adjacent faces on the 2D plane.

**Panorama Generation.** Panoramic image generation [13, 34, 40, 44, 8, 6, 18, 46, 38, 30, 37, 15] aims to synthesize 360-degree immersive images based on user-provided textual or visual clues. It requires the generated images to be seamless in spherical space while maintaining rationality in any perspective anchor. Recent studies [13, 34, 40, 44, 46, 38, 37, 15] leverage powerful image diffusion models to generate high-quality panoramic images. Benefiting from the great generative capabilities of diffusion models, these works have made significant advancements in omnidirectional image synthesis. Among them, PanoGen [15] achieves high-quality results in indoor scenarios by introducing a recursive outpainting and stitching mechanism. PanFusion [46] proposes a dual-branch pipeline to integrate equirectangular features and perspective features. CubeDiff [13] uses an alternative representation, cubemap, to synthesize panoramic images with high fidelity and diversity. Despite the significant progress in omnidirectional image generation, achieving similar success in the field of video remains a challenging problem, as it requires spatial-temporal consistency across the entire spherical space, as well as costly computational resources.

A few works [39, 16, 25, 33] explore panoramic video generation. 4K4DGEN [16] animates a given high-resolution panoramic image through user intention. However, the requirement of high-quality panoramic inputs limits the application scenarios, as it struggles to generate omnidirectional content without the given images. VidPanos [25] extends a given video to a larger FoV by using Temporal Coarse-to-Fine and Spatial Aggregation strategies, but it does not produce $360° \times 180°$ FoV panoramas. 360DVD [39] trains a 360-Adapter to exploit the generative capabilities of text-to-video diffusion models, achieving text-driven 360-degree video generation. Most relevant to our work, Imagine360 [33] utilizes a dual-branch design, similar to [46], to synthesize panoramic videos from given perspective inputs. However, the equirectangular videos generated by both 360DVD [39] and Imagine360 [33] exhibit severe distortion, especially at the poles, significantly degrading the sense of immersion and realism. We argue that the spatial distortion and motion drifting caused by ERP make it difficult for models to effectively understand the polar regions with limited training data. After all, generating perspective videos with extreme motion is still a challenge, let alone panoramas.

Current video generation models employ 3D positional encodings, enabling them to uniformly model videos across both spatial and temporal dimensions. Building on this foundation, open-source video models, such as Wan 2.1 [35] and CogVideoX [43], have acquired rich generative capabilities by training on extensive datasets of perspective videos. Multi-view generation approaches represent a panoramic video as a collection of multiple perspective videos. Although these perspective views align well with the native generative capabilities of video models, the spatial relationships between them are only inferred implicitly. This is typically achieved by introducing extra positional encodings or by constructing partial overlaps between adjacent faces. Such methods, however, tend to disrupt the model's inherent generative abilities and introduce artifacts like color discrepancies and visible seams in the resulting panoramic videos. Consequently, they fail to meet the demand for high-quality panoramic video generation. In contrast, an intact panoramic representation can naturally align with the native 3D attention mechanism of DiTs. In this paradigm, the model is able to capture global information across the entire panoramic scene, leading to the generation of superior panoramic videos. Our proposed ViewPoint Map is precisely such an explicit panoramic representation. It projects the non-Euclidean spherical data onto a single, unified Euclidean plane. This provides global continuity and allows the model to "see" the entire spherical space in a single pass.

**Video Outpainting.** Formally, extending from a given perspective view to a panoramic video is a kind of video outpainting task. Benefiting from pretrained diffusion models [10, 31, 29, 9], previous works [36, 5] have made notable progress in perspective video outpainting. However, due to the scarcity of 360-degree video data and the modality gap introduced by its unique representation form, panoramic video outpainting remains an open question. In this paper, we design a novel panorama

representation format to fully exploit the generative priors of video diffusion models, aiming to address the absence of high-quality panoramic video outpainting.

## 3 Method

### 3.1 Preliminary

Latent diffusion models [29, 10, 26] conduct a series of diffusion and denoising processes in latent space. Given a clean latent code $x_0$ from the training data, a noisy latent $x_t$ is obtained by adding an random noise $x_0 \sim \mathcal{N}(0, I)$ to $x_0$ according to a timestep $t \in [0, 1]$. Following the flow matching paradigm [17, 7], $x_t$ is defined as:

$$x_t = tx_1 + (1 - t)x_0. \tag{1}$$

The training objective of the model is to predict the velocity $v_t$, thus, the loss function of the training process can be formulated as:

$$L = \mathbb{E}_{x_0, x_1, c_{txt}, t} \left\| u(x_t, c_{txt}, t; \theta) - v_t \right\|^2, \tag{2}$$

where $c_{txt}$ is the text condition, $\theta$ is the model weights, $u$ is the predicted velocity and $v_t$ is the ground truth velocity.

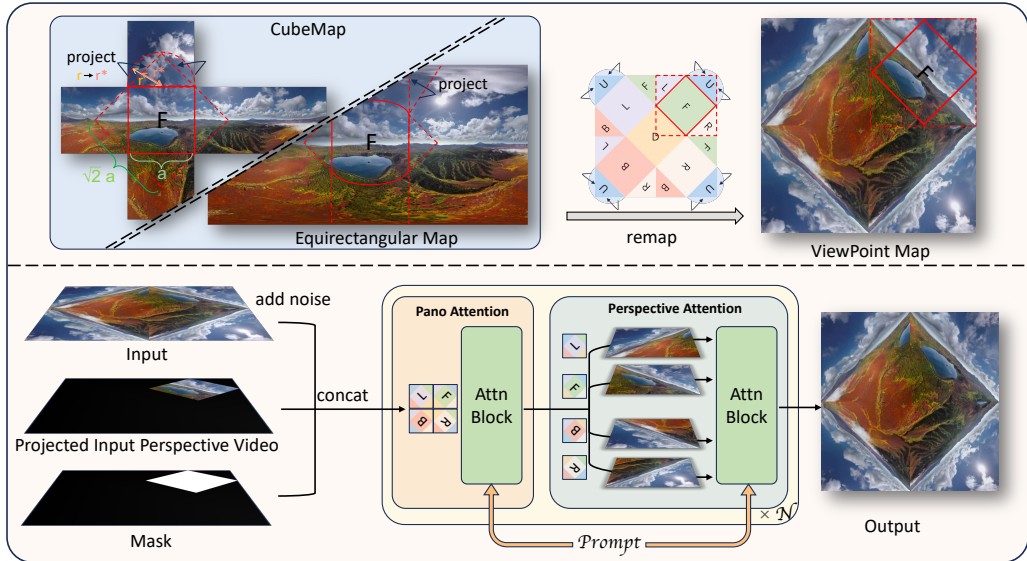

Figure 2: **Method overview.** The first row illustrates how to construct a Viewpoint map from a CubeMap or an Equirectangular Map. We begin by constructing subregions, using $\mathcal{F}$ as an example to create a pseudo-perspective region centered around it. Subsequently, we combine the four subregions to form a ViewPoint map. The second row shows the pipeline design: we first concatenate the noisy Viewpoint map, input video, and relevant mask along the channels dimension, and then employ a Pano-Perspective attention mechanism to learn how to maintain global spatial consistency while modeling fine-grained visual information.

### 3.2 ViewPoint Map

To maintain spatial continuity while reducing spatial distortion, we propose ViewPoint Map, a novel panorama representation that combines the advantages of ERP and CP. We first sample six cube faces from either an equirectangular map or a cubemap, namely $\mathcal{F}, \mathcal{R}, \mathcal{B}, \mathcal{L}, \mathcal{U}$, and $\mathcal{D}$, which represent the front, right, back, left, up, and down faces, respectively. The four faces in the horizontal view, i.e., $\mathcal{F}, \mathcal{R}, \mathcal{B}, \mathcal{L}$, typically contain most of the visual content that people focus on. Therefore, we construct pseudo-perspective subregions centered around these four faces. As shown in the first row

of Fig. 2, we select one face as the central face (assuming the side length of $a$), then diagonally dividing each of the four adjacent faces into four parts, and finally, the central face and its four adjacent parts are concatenated to form a square region with a side length of $\sqrt{2}a$. This design makes the central face more spatially continuous with the four adjacent faces. For example, if $\mathcal{F}$ is the central face, then its four adjacent faces are $\mathcal{L}, \mathcal{R}, \mathcal{U}$ and $\mathcal{D}$, located to the left, right, above, and below $\mathcal{L}$, respectively. After obtaining the four subregions, we perform rotation and concatenation on them to further ensure that the center of the ViewPoint Map—the $\mathcal{D}$ face—is also continuous. To address the splitting of $\mathcal{U}$, we project the semicircular region on the $\mathcal{U}$ face adjacent to the central face into an equilateral right triangle. Formulally, for any point $P(r, \theta)$ on the semicircular region, where $r \in [0, a]$ and $\theta \in [0, \pi]$, with $a$ being the diameter of the semicircle. The scale ratio is defined as:

$$d(\theta) = \frac{a}{sin\theta + |cos\theta|},$$

(3)

then the scaled radial coordinate $r^*$ is computed by:

$$r^* = r \cdot \frac{d(\theta)}{a}.$$

(4)

This design allows the $\mathcal{U}$ faces of the four subregions to have a small overlapping portion, thereby ensuring internal consistency of the entire $\mathcal{U}$ face.

### 3.3 Pano-Perspective Attention

ViewPoint Map offers global panorama information and effectively utilizes the inherent in-context generation capabilities [11] of diffusion models. To further accelerate convergence and exploit the priors of the base model, we propose a Pano-Perspective Attention mechanism. Specifically, Pano-Attention is responsible for contextual learning to maintain spatial consistency, while Perspective-Attention focuses on generating fine-grained visual information for each subregion. As shown in the second row of Fig. 2, we first concatenate the noisy Viewpoint Map, the projected input perspective video, and the associated mask along the channel dimension in the latent space. The shape of the concatenated features is $(batch\_size, channels, frames, height, width)$. After passing through the Pano Attention block, we reshape it to $(4 \times batch\_size, channels, frames, height/2, width/2)$ and feed it into the Perspective Attention for per-regional modeling. Each attention block consists of a self-attention layer, a cross-attention layer, and an FFN, the input prompt is integrated through the cross-attention mechanism.

### 3.4 Overlapping Fusion

Each subregion in a ViewPoint map has a partial spatial overlap with the two adjacent subregions. To achieve smoother transitions between subregions, we propose a gradient fusion mechanism. As shown in Fig. 3, the overlapping parts between two subregions form a regular rhombus, therefore we construct an overlapping fusion weight $W \in \mathbb{R}^{n \times n}$ where the values decay from the area near the central face to the edges.

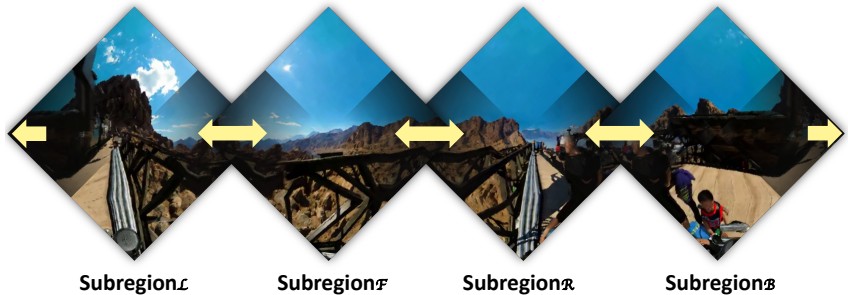

Subregion$_\mathcal{L}$     Subregion$_\mathcal{F}$     Subregion$_\mathcal{R}$     Subregion$_\mathcal{B}$

Figure 3: **Overlapping fusion.** The four subregions partially overlap with each other, thus we propose a gradient fusion mechanism to interpolate the overlapping areas, thereby enhancing spatial consistency.

For each subregion $S_d \in \mathbb{R}^{r \times r}$, $r = 2n$, $d \in [\mathcal{L}, \mathcal{F}, \mathcal{R}, \mathcal{B}]$, the overlapping fusion process can be described by the following formulas:

$$W_{i,j} = \frac{i+j-2}{2(n-1)}, \quad i,j = 1,2,\ldots,n \tag{5}$$

where $i, j$ denote the row and column indices, respectively. Next, we define rotation. For a given matrix $A$, we have:

$$R_{90}(A) = A^T \cdot J, \tag{6}$$
$$R_{-90}(A) = J \cdot A^T, \tag{7}$$
$$R_{180}(A) = J \cdot A \cdot J, \tag{8}$$

where $J$ is the exchange matrix. Finally, for each subregion, we have:

$$S_{L_{i,j}} = \begin{cases} S_{L_{i,j}} \cdot R_{90}(W) + R_{-90}(S_F)_{i+n,j-n} \cdot R_{-90}(W) \,, (i,j) \in [1,n] \times [n+1,r] \\ S_{L_{i,j}} \cdot R_{-90}(W) + R_{90}(S_B)_{i-n,j+n} \cdot Rot_{90}(W) \,, (i,j) \in [n+1,r] \times [1,n] \end{cases} \tag{9}$$

$$S_{R_{i,j}} = \begin{cases} S_{R_{i,j}} \cdot R_{90}(W) + R_{90}(S_F)_{i+n,j-n} \cdot R_{-90}(W) \,, (i,j) \in [1,n] \times [n+1,r] \\ S_{R_{i,j}} \cdot R_{-90}(W) + R_{-90}(S_B)_{i-n,j+n} \cdot R_{90}(W) \,, (i,j) \in [n+1,r] \times [1,n] \end{cases} \tag{10}$$

$$S_{F_{i,j}} = \begin{cases} S_{F_{i,j}} \cdot W + R_{90}(S_L)_{i+n,j+n} \cdot R_{180}(W) \,, (i,j) \in [1,n] \times [1,n] \\ S_{F_{i,j}} \cdot R_{180}(W) + R_{-90}(S_R)_{i-n,j-n} \cdot W \,, (i,j) \in [n+1,r] \times [n+1,r] \end{cases} \tag{11}$$

$$S_{B_{i,j}} = \begin{cases} S_{B_{i,j}} \cdot W + R_{-90}(S_L)_{i+n,j+n} \cdot R_{180}(W) \,, (i,j) \in [1,n] \times [1,n] \\ S_{B_{i,j}} \cdot R_{180}(W) + R_{90}(S_R)_{i-n,j-n} \cdot W \,, (i,j) \in [n+1,r] \times [n+1,r] \end{cases} \tag{12}$$

Note that we apply overlapping fusion to each subregion simultaneously, rather than in the order specified in the above formulas.

For the $\mathcal{U}$ face, we first project the triangles back into semicircles and transform the rhombus into a petal shape formed by the overlap of two circles. Finally, we use a similar fusion mechanism to fuse the overlapping of the four semicircles.

## 4 Experiments

### 4.1 Implementation Details

**Datasets & Preprocess.** Our model is trained on 4 panorama datasets, including one image dataset, Flickr360 [3], and three video datasets, WEB360 [39], ODV360 [3], and 360+x [4]. Among them, only WEB360 [39] comes with captions; therefore, we use Qwen-VL [2] to annotate the remaining three datasets, generating corresponding descriptive captions. All datasets are resized to a resolution of $512 \times 1024$, and during training, video data is divided into clips of 49 frames each.

**Training.** We first inflate the patch embedding layer of the powerful video generation model, Wan2.1 [35], from 16 to 33 channels to accommodate the input data, and then fine-tune the entire model. The training process is executed on $8 \times$ NVIDIA A100 GPUs, using a batch size of 1 and a learning rate of $1e-4$. We employ a joint image-video training strategy, treating images as videos with only one frame.

### 4.2 Qualitative Comparison

We compare our approach with three previous methods, including one perspective video outpainting method, Follow-Your-Canvas [5], and two 360-degree video generation methods, 360DVD [39] and Imagine360 [33]. Adopting equirectangular representation, both 360DVD [39] and Imagine360 [33] exhibit severe distortion at the poles. As shown in Fig. 4, the sky appears to have "black holes", while the ground shows "swirls". We argue that these artifacts are caused by the model's difficulty in handling the spatial-temporal distortion introduced by ERP, as a slight movement can result in significant disturbances at the poles.

The visualization of the comparison results is shown Fig. 5, where we present the ERP format and perspective views in four horizontal directions, with arrows indicating the flow of time. As shown in Fig. 5, 360DVD [39] only recognizes textual input. The generated frames show limited

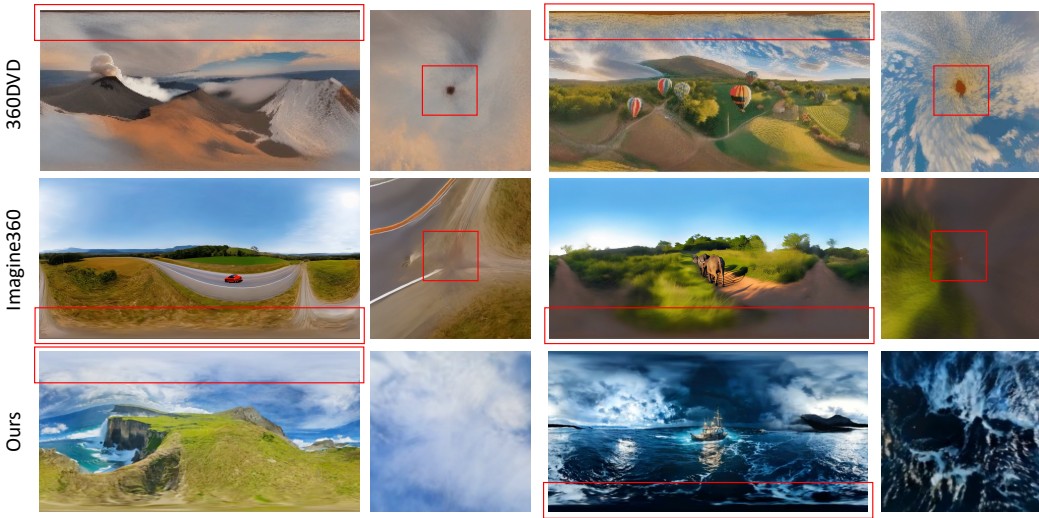

Figure 4: **Distortion in 360DVD [39] and Imagine360 [33].** Despite numerous efforts to mitigate geometric distortion at the poles, both methods still struggle with generating realistic top and bottom views. The distortion at the poles is even more pronounced in video scenes, severely affecting realism and immersion.

visual quality and exhibit minimal motion. Follow-Your-Canvas (F-Y-C) [5] generates satisfactory perspective videos; however, its results suffer from severe spatial inconsistency, especially on the $\mathcal{B}$ face, exhibiting a noticeable sense of disconnection. Although Imagine360 [33] synthesizes reasonable panoramic videos, it lacks spatial-temporal continuity. As in the first example, the little girl is running, but the ground does not move accordingly and shows a noticeable texture difference from the input video. In the second example, the aircraft in the input video is in a dark and tense atmosphere, while the generated panoramic video shows blue skies and white clouds, creating a peaceful scene. Additionally, the aircraft does not integrate well with the surrounding environment, resulting in a discordant effect. In contrast, our approach demonstrates excellent spatial-temporal consistency and supports both textual prompts and input videos, thus enhancing the flexibility of generation. Furthermore, ViewPoint exhibits high motion dynamics, as each perspective view responds to the motion trends of the input video correctly. As in the first case, the input video depicts a little girl running forward, with the prompt describing a magical forest scene at night. Our generated video aligns with both the input video and the prompt, showcasing a strong capability for condition awareness and integration. Similarly, in the second example, the generated panoramic video also integrates with the input video and exhibits high-quality dynamics.

### 4.3 Quantitative Comparison

In this section, we provide a quantitative comparison of our approach with previous methods. VBench [12] is a comprehensive benchmark suite for video generative models which scores a video in 16 dimensions. We evaluate our approach and previous methods on the ODV360 [3] dataset across five dimensions: "subject consistency", "imaging quality", "motion smoothness" and "dynamic degree". Specifically, we first project the generated videos into six perspective views with a FoV of 90°, similar to $\mathcal{F}, \mathcal{R}, \mathcal{B}, \mathcal{L}, \mathcal{U}$, and $\mathcal{D}$. Then, we apply VBench [12] evaluation in a perspective manner for all projected videos.

The results of the quantitative evaluation are shown in Tab. 1, where our approach achieves the best scores across four metrics. Although 360DVD [39] performs the second best in "subject consistency", it scores the lowest in "dynamic degree", indicating that the videos generated by 360DVD have very little motion. The nearly static videos result in 360DVD achieving a high score in "motion smoothness"; however, such videos fail to meet the demands of generating high-quality panoramic videos. Follow-Your-Canvas [5], on the other hand, is capable of generating high-quality perspective results, but the projected videos exhibit severe spatial distortion, resulting in poor performance in evaluations. In contrast, our approach demonstrates state-of-the-art performance in all four metrics, proving that our method can generate highly dynamic and temporally coherent panoramic videos.

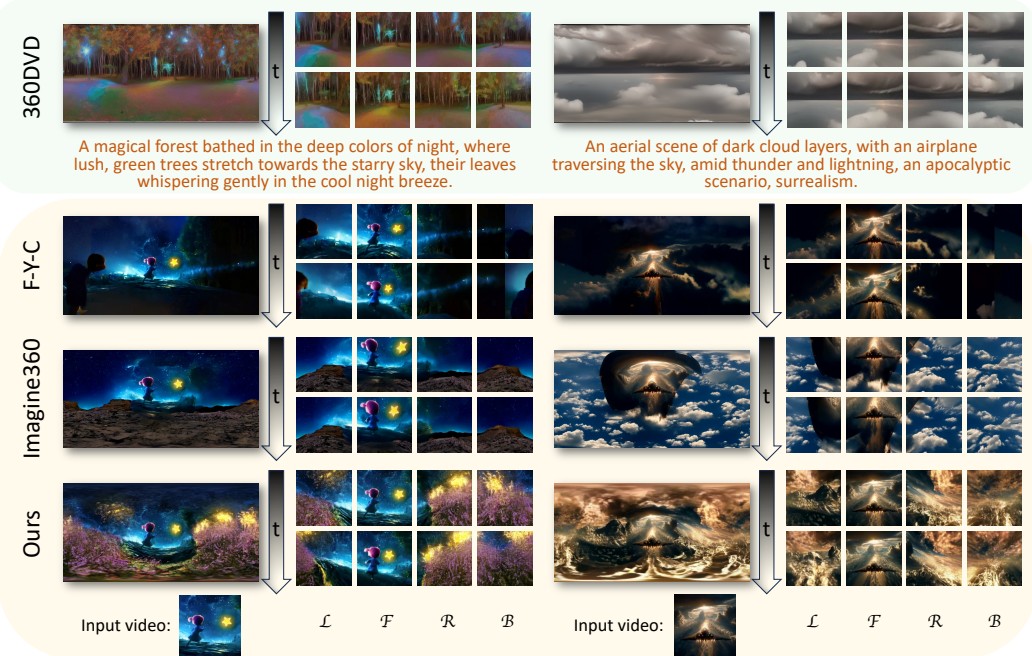

Figure 5: **Qualitative comparison of generated videos.** 360DVD is a text-driven approach and produces results with limited visual quality while Follow-Your-Canvas fails to generate panoramic videos with a reasonable spatial layout. Imagine360 suffers from spatial-temporal discontinuity. Our approach, on the other hand, can generate high-quality panoramic videos aligned with the given input. Due to page limits, we highly recommend watching the dynamic videos available at ViewPoint.

Table 1: **Quantitative comparison on VBench.** Our method achieves the best performance in all four metrics. Although 360DVD obtains the second highest "subject consistency" score due to the nearly static nature of its generated videos, the results fail to meet the demands of generating high-quality panoramic videos. In contrast, our approach performs well in both motion consistency and dynamics.

| Method | subject consistency↑ | imaging quality↑ | motion smoothness↑ | dynamic degree↑ |
|---|---|---|---|---|
| Follow-Your-Canvas [5] | 0.8284 | 0.4464 | 0.9655 | 0.8500 |
| 360DVD [39] | 0.8633 | 0.5394 | 0.9703 | 0.5083 |
| Imagine360 [33] | 0.8547 | 0.5859 | 0.9720 | 0.8148 |
| ViewPoint(AnimateDiff [9]) | 0.8681 | 0.5914 | 0.9786 | 0.8633 |
| **ViewPoint(Wan2.1** [35]) | **0.8793** | **0.5927** | **0.9800** | **0.9083** |

## 4.4 Ablation Study

To demonstrate the effectiveness of our method, we conduct ablation experiments on panorama representation formats and network designs. For the ERP format, we directly fine-tune the base model to adapt to this panoramic representation. For the CP format, we train two models, CP-ICLoRA and CP-4DRoPE, separately to assess the impact of different cubeface-encodings. Technically, CP-ICLoRA rearranges the six cube faces into a 2-row by 3-column rectangle in raster order, *i.e.*, $\mathcal{F}, \mathcal{R}, \mathcal{B}$ on the first row, and $\mathcal{L}, \mathcal{U}, \mathcal{D}$ on the second row, while CP-4DRoPE assigns an independent positional encoding to each cube face. We also examine the design of Pano-Perspective attention by replacing all Perspective blocks with the intact Pano blocks.

As shown in Fig. 6, ERP exhibits significant artifacts due to the inherent modality gap with the pretrained model. Both cubemap representation methods suffer from spatial discontinuity, wherein CP-4DRoPE shows inconsistency in color tone, while CP-ICLoRA, although achieving better image quality, presents significant spatial fractures. Our full method, in contrast, is capable of generating high-quality and coherent panoramic videos that are superior to other alternative designs in both spatial and temporal aspects.

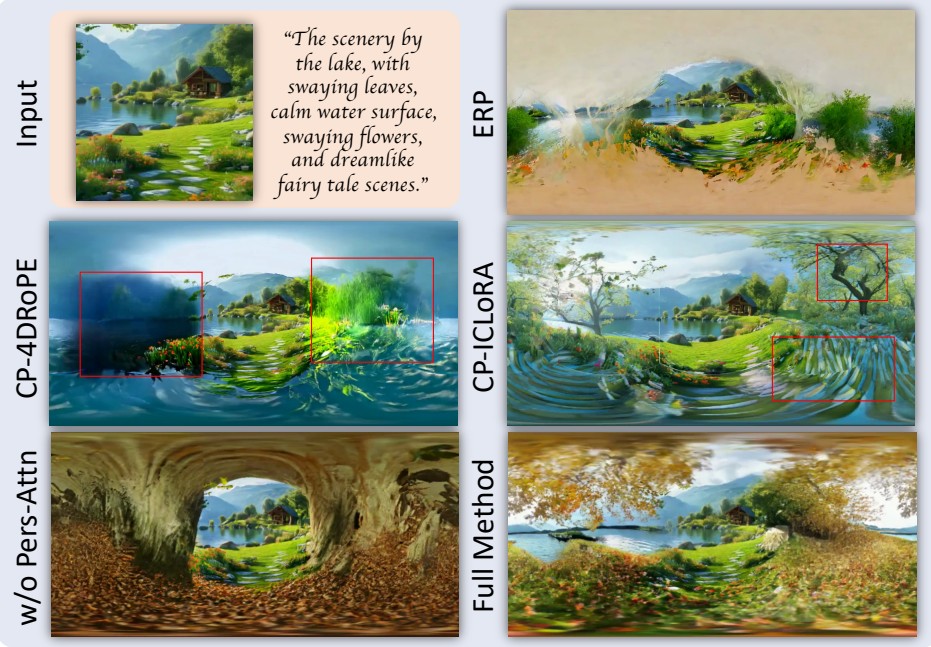

Figure 6: **Ablation on different designs.** ERP exhibits serious artifacts due to the natural gap in modality. Both Cube representation methods have spatial discontinuity issues. Without Perspective-Attention, it leads to misalignment with the input video. Our full method can generate reasonable and spatially consistent results.

The results of quantitative ablation are shown in Tab. 2, our full method outperforms the other four alternative designs across four metrics. Since we conduct evaluations in a perspective manner, CP-ICLoRA performs best in "imaging quality". However, as shown in Fig. 6, though each perspective video shows good visual quality, there is still a discontinuity issue at the boundaries of the cube faces.

Table 2: **Quantitative ablation.** Compared to the other four alternative designs, our approach achieves the highest scores across three metrics. Although CP-ICLoRA achieves the highest score in "imaging quality", proving that each of the six faces has good quality independently, the panoramas composed of these six faces exhibit noticeable discontinuities, as indicated by the red boxes in Fig. 6. Therefore, it does not meet the demand for high-quality panoramic video generation.

| | subject consistency↑ | imaging quality↑ | motion smoothness↑ | dynamic degree↑ |
|---|---|---|---|---|
| ERP | 0.8702 | 0.5396 | 0.9781 | 0.8840 |
| CP-4DRoPE | 0.8612 | 0.5104 | 0.9633 | 0.8175 |
| CP-ICLoRA | 0.8727 | **0.6018** | 0.9786 | 0.8946 |
| w/o Perspective-Attn | 0.8663 | 0.5392 | 0.9798 | 0.8827 |
| **Full Method** | **0.8793** | 0.5927 | **0.9800** | **0.9083** |

To further validate the effectiveness of our design, we conduct a user evaluation with 50 participants to compare different designs across four dimensions: "Spatial Continuity", "Temporal Quality", "Aesthetic Preference" and "Condition Alignment" (detailed in section 4.5). As shown in Tab. 3, the results demonstrate a significant user preference for our method over the alternatives.

Table 3: **User evaluation.** Compared to the other designs, users show a significant preference for our method across all four dimensions.

| | Spatial Continuity | Temporal Quality | Aesthetic Preference | Condition Alignment |
|---|---|---|---|---|
| ERP | 3(6%) | 3(6%) | 2(4%) | 4(8%) |
| CP-4DRoPE | 3(6%) | 2(4%) | 3(6%) | 2(4%) |
| CP-ICLoRA | 2(4%) | 4(8%) | 11(22%) | 7(14%) |
| w/o Perspective-Attn | 11(22%) | 9(18%) | 7(14%) | 12(24%) |
| **Full Method** | **31(62%)** | **32(64%)** | **27(54%)** | **25(50%)** |

### 4.5  User Study

Despite achieving advanced scores on VBench [12], we conduct user studies, introducing subjective user ratings to further validate the superiority of our approach. We ask participants to vote on the tested videos based on four dimensions: "Spatial Continuity", "Temporal Quality", "Aesthetic Preference" and "Condition Alignment". "Spatial continuity" represents the coherence of scenes and structures in panoramic space, while "Temporal Quality" is used to evaluate motion effects; for example, being nearly motionless and flickering are considered poor performance. "Aesthetic Preference" represents participants' subjective preferences, while "Condition Alignment" refers to the degree to which the generated panoramic video aligns with the input conditions, such as text or video.

Finally, we collect 50 valid questionnaires, each containing 14 sets of videos for comparison, and the results are shown in Fig. 7. Overall, Imagine360 [33] is the second choice among participants, while 360DVD [39] and Follow-Your-Canvas [5], receive the fewest votes. Our approach receives the highest number of votes across all four dimensions, surpassing 360DVD [39], Follow-Your-Canvas [5], and Imagine360 [33], demonstrating our method's ability to generate satisfactory panoramic results.

## 5  Conclusion

In this work, we present ViewPoint, a novel framework for representing and generating panoramic videos leveraging modern generative models. Specifically, we design a novel representation distinct from traditional panoramic image and video representations, which has the advantage of good spatial continuity and temporal consistency. Through our proposed Pano-Perspective attention mechanism, the pretrained model perceives the global spatial structure information of panoramic videos while modeling fine-grained visual features, effectively improving the quality of the generated videos. To further enhance spatial consistency, we propose an overlapping gradient fusion mechanism that fully utilizes the spatial continuity of each subregion, thereby improving the spatial quality of panoramic videos. Extensive qualitative and quantitative experiments, as well as user studies, demonstrate the effectiveness of our method, and we believe ViewPoint can provide valuable insights for the omnidirectional vision community.

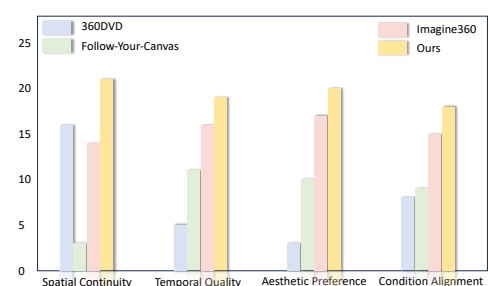

Figure 7: **User studies.** We ask participants to vote on the videos generated by four methods based on four dimensions, and our approach receives the most votes.

## 6  Limitations and Future Work

A significant challenge in panoramic video generation is achieving high resolution. Immersive applications such as virtual reality (VR) necessitate ultra-high resolutions (e.g., 8K, 16K) to deliver a realistic and compelling user experience. However, current pre-trained diffusion models are computationally constrained and cannot directly synthesize content at such scales. Consequently, in line with established practices, our work employs a two-stage generate-then-upscale pipeline to mitigate this limitation.

Looking ahead, our future work will focus on two primary directions. First, we aim to explore end-to-end models for ultra-high-resolution panoramic generation. Second, we will investigate the extension to long-form video generation, for instance, by leveraging autoregressive frameworks.

### Acknowledgments

This work is supported by the National Natural Science Foundation of China (NSFC) under Grants 62576328, 62306295, 62225207 and 62436008.

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

# Appendix

In this appendix, we discuss the usage of computational resources, including VRAM consumption and inference latency in section 7. In section 8, we demonstrate that our method can support text-driven panoramic video generation, further expanding its application scenarios. Afterwards, we present the limitations of our method in section 9. Finally, we discuss the potential societal impact in section 10.

## 7 Computational Resources

### 7.1 Representation Comparison

We compare our method with two mainstream panorama representations in terms of area. As shown in Fig. 8, assuming that the side length of a perspective anchor with a field-of-view (FoV) of 90°×90° is r, the area of the equirectangular representation is 8r², and the area of the 2D grid in the cubemap is 12r², among which the valid representation area is 6r². In contrast, our representation occupies an area of 8r².

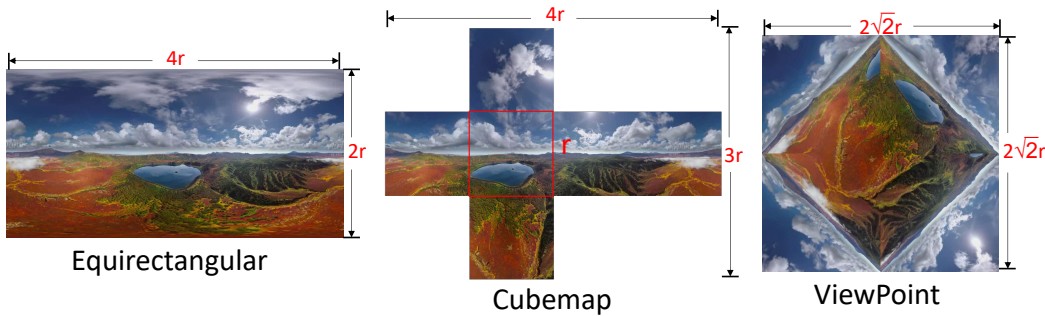

Figure 8: Comparison of different representations in terms of area.

### 7.2 Methods Comparison

We compare our method with 360DVD [38], Follow-Your-Canvas [5], and Imagine360 [33] in terms of model complexity and inference-time latency on a single NVIDIA A800(80GB). As shown in Tab. 4, in the case of generating $512 \times 1024$ 49-frame equirectangular videos (with the exception of 360DVD, which is limited to 16-frame video generation), our method demonstrates superior performance in terms of inference latency, model parameters, and peak VRAM consumption compared to prior approaches. Note that we follow the original settings of the compared methods. Among them, Follow-Your-Canvas [5] exhibits extremely long inference latency due to the progressive outpainting design. Imagine360 [33], on the other hand, shows significantly high model complexity and a large number of parameters, resulting from its dual-branch architecture. In contrast, our method achieves excellent model efficiency, generating high-quality results with significantly fewer computational resources.

Table 4: Comparison of computational resource usage with previous methods. Note that 360DVD only supports the generation of 16-frame videos, while the other three methods are evaluated in the 49-frame scenario.

|  | 360DVD(16 frames) [39] | Follow-Your-Canvas [5] | Imagine360 [33] | Ours |
|---|---|---|---|---|
| Latency↓ | 1min30s | 29min19s | 4min44s | 1min47s |
| Peak VRAM↓ | 17.44GB | 32.57GB | 40.45GB | 20.77GB |
| Parameters↓ | 1.33B | 1.42B | 5.94B | 1.42B |

## 8 More Application

Our model not only supports video input, but also pure text input. Therefore, our method can also be applied to text-driven panoramic video generation. As shown in Fig. 9, our approach is capable of generating high-quality panoramic video using only textual guidance.

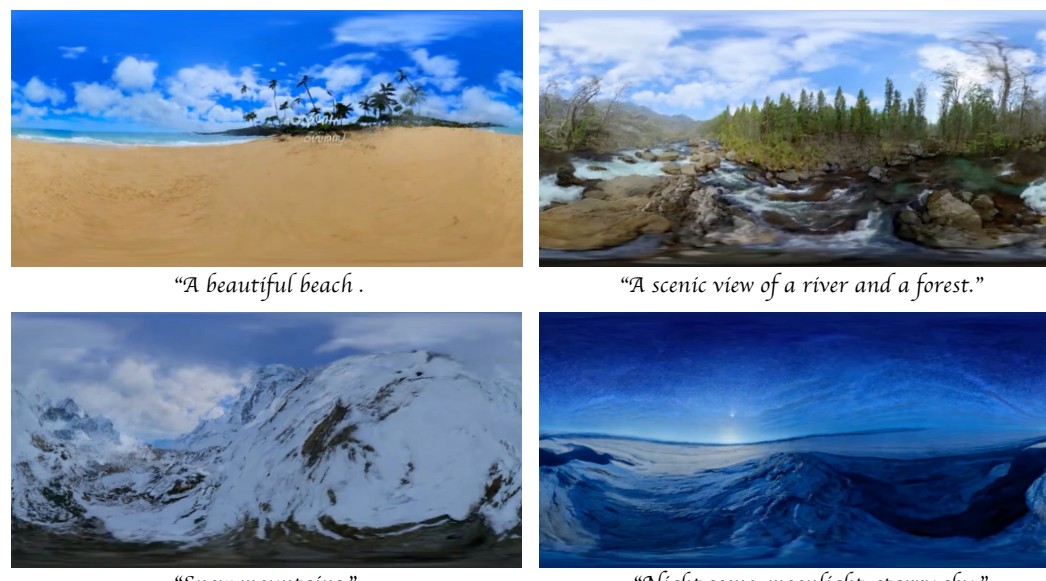

*"A beautiful beach .*          *"A scenic view of a river and a forest."*

*"Snow mountains."*          *"Night scene, moonlight, starry sky."*

Figure 9: Text-driven generation. Our method can generate high-quality panoramic video using only textual guidance.

## 9 Limitations

As shown in Fig. 10, the panoramic videos generated by our model contain watermarks, exhibit mosaic artifacts on human faces, and occasionally reveal photographic equipment such as selfie sticks in the $\mathcal{D}$ regions of the video, which can degrade the immersive experience. These limitations can be attributed to the presence of watermarks and specialized post-processing in our training data [4, 39]. Specifically, 360 camera manufacturers typically embed their logos into the recorded videos, leading to an abundance of watermarked content within the dataset. In addition, some works [4] choose to apply a mosaic effect to human faces in the datasets for ethical and moral considerations. We acknowledge and respect the ethical considerations taken by the creators of the datasets. We argue that these limitations can be addressed by using higher-quality datasets, where high quality refers to, for example, removing watermarks or hiding photographic equipment through algorithms, as well as filtering out video clips containing human faces.

## 10 Potential Societal Impact

Panoramic video generation has the potential to bring significant positive societal impact. It can empower non-expert users to create high-quality immersive content, thereby fostering innovation in digital art, virtual tourism, education, and cultural heritage preservation. This technology also enhance remote experiences and contributes to accessibility, enabling broader participation in virtual environments for individuals with physical or economic limitations. Furthermore, it offers valuable support in scientific visualization and educational applications, especially in fields such as geography and urban planning. However, generating virtual panoramas also pose several challenges. Concerns around privacy and ethics arise when real-world data containing personal identities are used for training. The potential for misuse, including the creation of deepfakes and misleading visual content, raises serious issues regarding misinformation and public trust. Additionally, copyright infringement

and labor displacement in creative industries are growing concerns. To mitigate these risks, it is essential to implement robust regulatory frameworks, ethical guidelines, and public awareness campaigns that promote responsible use and transparency in AI-driven content generation.

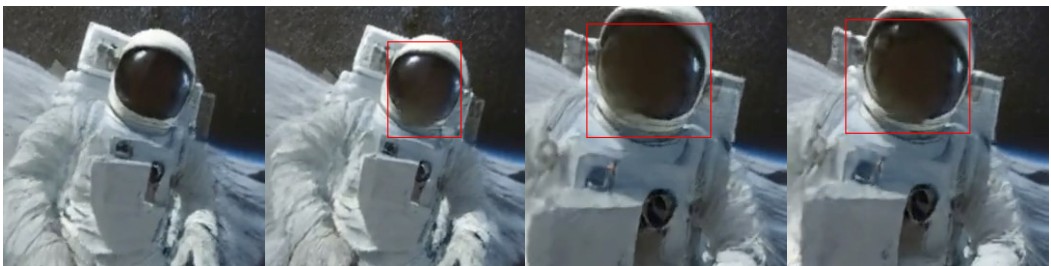

Mosaic effect

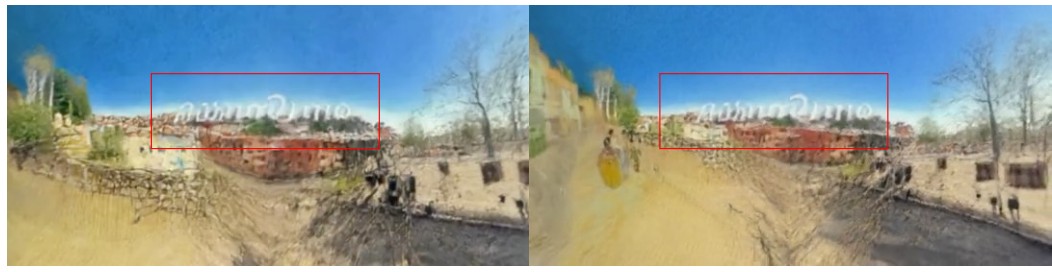

Watermarks

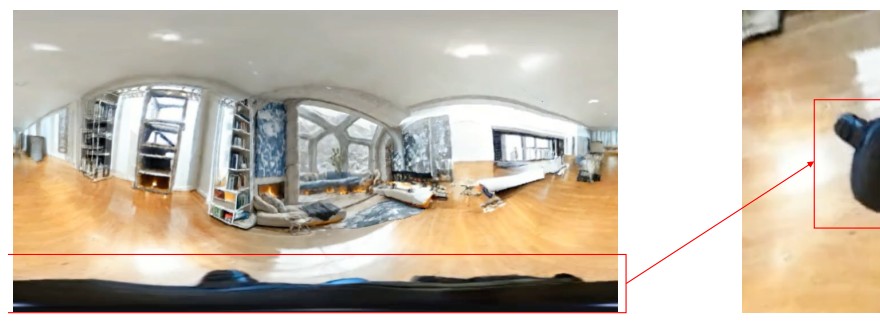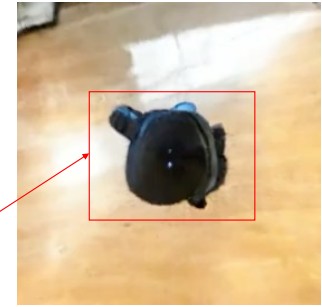

Photographic equipment

Figure 10: Potential visual artifacts.

