# OpenReview forum: "ViewPoint: Panoramic Video Generation with Pretrained Diffusion Models"
_NeurIPS.cc/2025/Conference — NeurIPS 2025 poster_

### Official Review · Reviewer_dty6 · 2025-07-03

**Clarity:** 2
**Significance:** 2
**Originality:** 3
**Rating:** 4
**Confidence:** 4

**Summary:**

The paper proposes a method for adapting a text-to-video diffusion model to upconvert narrow field-of-view videos into seamless 360 panoramic videos. The two main contributions are a parameterization of the panorama into the square domain of the diffusion model that reduces perspective distortion, and a modification to the diffusion model attention mechanism and optimization that allows seamless blending across the edges of the square domain.

**Questions:**

Could the method incorporate camera stabilization by adjusting the mask and input perspective view? If not, what is the advantage here over directly generating a 360 video?

Can the ERP results in Fig. 6 be improved with more prompt engineering?

In 4.1, how is a single image fed into the model through the TAE? Is it just stacked?

I'm not clear on exactly what is happening in 3.4, the overlapping fusion section. Is this basically multidiffusion (Bar-Tal, et al) with some rotation and translation?

**Ethical Concerns:**

["NO or VERY MINOR ethics concerns only"]

**Final Justification:**

I have read the authors' response and comments. Post discussion I am willing to increase my rating given the authors claim to include results on real videos. Please make sure that if the paper is accepted these results are added to the paper.

**Limitations:**

yes

**Quality:**

3

**Strengths And Weaknesses:**

The method effectively addresses the issue of seamless 360 panorama generation for a video diffusion model. The presented results do not show visible seams or singularities, though there are some quality and resolution issues, possibly due to the base Wan 2.1 model. Existing video-generation models can create 360 videos when prompted with "360", however since they do not explicitly handle stitching they tend to show seams and singularities (search for "veo 3 360 videos" on youtube). The method itself is simple and reasonable: it effectively unwraps a cubemap projection into a square domain, creating triangular flaps to handle overlap.

While I believe that straight generation of 360 videos with an unmodified Wan 2.1 would show seams, it would still have been nice to show some baseline results simply by prompting a Wan 2.1 outpainting model with "360" or some variation of that prompt. Given the Veo results, I would expect it would be able to produce at least an approximation of an equirect video. Fig. 6 suggests that there is a domain gap that causes bad performance for ERP, but without some more exploration of the prompt I'm not sure I believe this is true.

The paper doesn't mention camera stabilization, so I assume the input perspective view is always fed into the "F" cubemap panel (Fig. 2). Compared to previous work like VidPanos, the problem being solved here is slightly different: since the camera is not stabilized, there is not an expectation that content outside the current input perspective view is kept consistent across the generation. This feels like a limitation if the method is applied to real captured videos, but if the target is extension for generated videos, why not just generate the entire 360 video in one shot? It would be interesting to see what the method does on the VidPanos dataset, though as the paper states vidpanos is not a full 360 dataset or method.

While the perspective and panorama attention modifications are useful for reducing seams, they also feel like a practical limitation as they specialize the model architecture to this particular 360 parameterization. It also raises the question of whether using a modified attention mechanism could solve the issues with the ERP projection, without using the presented projection. It is likely that singularities above and below would still exist, but a wrapping attention mechanism might address seams.

The language in a few places should be tweaked. It's not necessary to say competing methods "exhibits terrible image quality" (Fig. 5 caption), for example; this is in the end subjective and unnecessarily negative.

---

> ### Author Rebuttal · Authors · 2025-07-30
>
> Dear Reviewer dty6, we would like to express our sincere gratitude for your insightful and thoughtful feedback. We are also grateful for your positive comments on the quality of our results. We have carefully considered your questions, and our responses are as follows:
> ### **Q1: The Effect of Prompt Engineering.**
> Prompt engineering experiments on the base model (Wan2.1-1.3B) fail to produce ERP-like videos, confirming a significant modality gap with the ERP representation.
>
> In these experiments, we use phrases like "a 360 video of ..." to denote panoramic formats. Due to rebuttal guidelines, we cannot include external links or update the anonymous link in the paper. While we regret being unable to showcase the video results, we will describe the observed phenomena in detail below.
>
> Since Wan2.1 only supports a fixed aspect ratio (e.g., 832x480), which does not conform to the 2:1 aspect ratio of the ERP format, we divide our experiments into two settings:
>
> 1. Using the model's native, fixed aspect ratio.
> > Observation: The generated results do not exhibit ERP-like characteristics. Instead, they resemble a bird's-eye perspective. Furthermore, both the visual quality and the dynamism are significantly degraded, with the video being nearly static.
>
> 2. Adjusting the aspect ratio to 2:1 (e.g., 960x480) to accommodate the ERP format.
> > Observation: After adjusting the aspect ratio to 2:1, the visual quality improves slightly, but the dynamism remains unchanged, and the output still does not resemble ERP format. Instead, it resembles a wide FoV perspective view, like a landscape wallpaper. (Again, we regret being unable to directly showcase the results.)
>
> Regarding the Veo3 360 videos on YouTube, we have reviewed the content and agree that the results are impressive. However, creators note that they require extensive post-processing using professional software like Premiere.
>
> Our analysis suggests that Wan2.1's failure to generate ERP-like videos stems from a modality gap introduced during pre-training, due to data preprocessing steps like cropping or resizing, which damage the spatial structure ERP data depends on.
>
> To further investigate the impact of the prompt on the generation results, we design ERP-specific prompts in our ablation study. The results show no significant performance improvement, supporting our claim of a modality gap between Wan2.1 and the ERP format.
> | Method | subject consistency | imaging quality | motion smoothness | dynamic degree |
> | --- | :---: | :---: | :---: | :---: |
> | ERP+"a 360 video of..." | 0.8693 | 0.5412 | 0.9770 | 0.8829 |
> | ERP+"a panoramic video of..." | 0.8711 | 0.5391 | 0.9776 | 0.8832 |
> | ERP (original) | 0.8702 | 0.5396 | 0.9781 | 0.8840 |
>
> ### **Q2: Camera Stabilization and Experiments on the VidPanos Dataset.**
> During training, we simulate camera movement by randomly perturbing the Euler angles, which enables our method to generalize flexibly to various input scenarios. This endows our method with a built-in camera stabilization capability, making it applicable to real-world scenarios. Intuitively, this process is equivalent to rotating each frame of the spherical video and then sampling the perspective view from a fixed anchor. We provide the details of the process below:
>
> Since all training data is in ERP format—the most common for panoramic data—a specific preprocessing step is required. We extract a perspective video from the ERP as the condition and convert the full ERP into our ViewPoint Map representation (as shown in Fig. 2). Specifically, the conditional video is obtained through the following steps:
>
> 1. We randomly generate a sequence of continuously varying Euler angles, where the length of the sequence is identical to that of the ERP video. The angles are constrained to the following ranges: roll is within (-10°, 10°), pitch is within (-30°, 30°), and yaw is within (0°, 360°). For instance, for a 5-frame video, we would obtain three distinct sequences for the Euler angles, such as:
> ***roll***: [0°, 1°, 0°, -1°, -2°]; ***pitch***: [4°, 2°, 1°, -1°, 2°]; ***yaw***: [20°, 23°, 22°, 20°, 18°]
> 2. Each set of Euler angles is then applied to its corresponding frame in the ERP video to perform a rotation. That is, the first frame is rotated using the angles {***roll***: 0°, ***pitch***: 4°, ***yaw***: 20°}, the second frame is rotated using {***roll***: 1°, ***pitch***: 2°, ***yaw***: 23°}, and so on.
> 3. From this rotated ERP video, we project the conditional perspective video using a fixed anchor (e.g., F). Finally, the entire rotated ERP is converted into a ViewPoint Map for training.
>
> This simple yet effective data augmentation strategy enables our model to perform camera stabilization without needing to explicitly estimate camera parameters from the input perspective video.
>
> To validate our design, following your suggestion, we conduct qualitative and quantitative evaluations on **VidPanos-real**. The results confirm the superiority of our method. Although we cannot provide new visualizations in this rebuttal, relevant real-world examples are always available via the anonymous link in our paper. For instance, in the "red car" case, while the INPUT VIDEO shows part of the car moving out of frame, the PANO VIDEO retains the full body, supporting our qualitative claims. Quantitative results are shown in the table below.
>
> | Method | subject consistency | imaging quality | motion smoothness | dynamic degree |
> | --- | :---: | :---: | :---: | :---: |
> | Imagine360 | 0.8511 | 0.5694 | 0.9716 | 0.8365 |
> | ViewPoint | 0.8780 | 0.5872 | 0.9729 | 0.9132 |
>
> The results demonstrate the effectiveness of our method. Unlike Imagine360, which uses external models [1,2] to estimate camera parameters and generate panoramic video, our approach directly embeds camera movement into the model, avoiding potential error accumulation.
>
> [1]A. Veicht, *et al*. GeoCalib: Single-image calibration with geometric optimization. ECCV,2024.
>
> [2]L. Jin, *et al*. Perspective fields for single image camera calibration. CVPR,2023.
>
> ### **Q3: Explanation of the Pano-Perspective attention mechanism.**
> Although using the ERP representation combined with a well-designed attention mechanism may also help alleviate the seam issue, our approach goes beyond by not only addressing the seam problem but also mitigating the polar distortion and more effectively leveraging the generative capabilities of pre-trained video models, thus enabling the generation of high-quality panoramic videos.
>
> The method you mentioned can be referenced to Imagine360, which projects ERP into multiple perspectives and employs a cross-domain spherical attention mechanism in an attempt to bridge the modality gap between panoramic data and the pre-training data of video models. While this approach is able to address the seam issue, the polar distortion remains severe (as shown in Figure 4). Moreover, it fails to fully exploit the generative power of existing video models, resulting in lower-quality panoramic video outputs (e.g., limited motion, see Table 1 in the main paper).
>
> In contrast, our representation, together with the proposed Pano-Pers attention mechanism, enables global structural modeling of panoramic data and further exploits the generative capacity of video models through the construction of pseudo-perspective sub-regions. This capability is not achievable by other representations or attention mechanisms.
>
> ### **Q4: Wording and Phrasing.**
> Thank you for pointing out our inappropriate wording.
> We have revised line 192-193, changing the sentence
> > "The generated frames exhibit terrible image quality and possess little motion."
>
> to
>
> > "The generated frames show limited visual quality and exhibit minimal motion."
>
> and updated the caption of Figure 5 from
> > "360DVD is a text-driven approach and exhibits terrible image quality"
>
> to
>
> > "360DVD is a text-driven approach and produces results with limited visual quality"
>
> ### **Q5: How is a single image fed into the model?**
> Consistent with the prevailing training paradigm of video generation models (e.g., Wan2.1), we feed images into the model by treating them as single-frame videos, thanks to the image-video unified encoding VAE. Specifically, a video tensor with a shape of ***[batch, channels(RGB:3), frames, height, width]*** is first passed into the Temporal AutoEncoder of Wan 2.1. Given that the VAE has a temporal-spatial downsampling factor of ***(4, 8, 8)***, the input data must satisfy two conditions: the number of frames must be ***frames = 4t + 1*** (where t is a non-negative integer), and both height and width must be multiples of 8. After being encoded by the VAE, the video is transformed into a latent code with the shape ***[batch, channels(latent:16), t+1, height//8, width//8]***, which is then fed into the DiT for modeling. For image data, we treat it as a video with the shape ***[batch, channels(RGB:3), 1, height, width]***. After encoding, its shape becomes ***[batch, channels(latent:16), 1, height//8, width//8]***.
> ### **Q6: Details on Overlapping Fusion Operation.**
> Different from MultiDiffusion, which enforces overlapping consistency in the noise space, our proposed overlapping fusion operation is performed in the pixel space, thereby improving spatial continuity. Specifically, we first rotate the four sub-regions (i.e., L, F, R, B) derived from the ViewPoint Map and align them horizontally, as illustrated in Fig. 3. Subsequently, a weighted linear interpolation is performed in pixel space. As shown in Fig. 3, each gradient black diamond shape represents an interpolation weight matrix, where a darker color indicates a smaller weight, and a more transparent color (i.e., closer to the original pixel color of the sub-region) signifies a larger weight. Each subregion is interpolated with its two adjacent faces based on this weight matrix.

---

> > ### Comment · Reviewer_dty6 · 2025-08-05
> > **Q5 and Q6**
> >
> > Thanks for the extra information. A couple followups:
> >
> > Q5: minor, but where does the 4t + 1 come from if the TAE downsampling is (4,8,8)? Does the TAE take a special single frame input at the beginning and then chunks of 4 frames, similar to some other AE designs?
> >
> > Q6: I do not follow the distinction here between pixel space and noise space. Since the generator used here is an LDM, does this mean that the overlapping fusion is done post-generation and post-decoding, unlike multidiffusion, which fuses partial results during the sampling loop?

---

> > > ### Author Response · Authors · 2025-08-06
> > >
> > > Dear Reviewer dty6, we sincerely thank you for your reply. Our responses are as follows.
> > >
> > > **Q5-Answer**: Yes, you are right. Specifically, to accommodate joint video-image encoding, TAE does not perform temporal downsampling on the first frame, and therefore takes the input video with 4t + 1 frames (when t = 0, the input video has only one frame, i.e., an image), which fully inherits the design of SOTA open-source video generation models, such as Wan 2.1 and CogVideoX.
> > >
> > > **Q6-Answer**: Yes, you are totally correct. After the sampling loop, we decode the latent code back to the pixel space and then perform the overlapping fusion to further improve spatial smoothness.
> > >
> > > We will refine the descriptions of joint video-image encoding and overlapping fusion in the final version to provide a clearer explanation of the method for the readers.
> > >
> > > Once again, we sincerely thank you for your time and effort. If you have any further concerns, please do not hesitate to let us know—we would be more than happy to discuss them with you.

---

> > > ### Author Response · Authors · 2025-08-07
> > >
> > > Dear Reviewer dty6,
> > >
> > > As the discussion phase is approaching its end, we sincerely would like to ask if you have any remaining concerns? Your valuable feedback and insightful comments are of great importance to us. We truly appreciate the time and effort you have devoted, and we extend our most sincere gratitude to you.
> > >
> > > Best regards,
> > >
> > > ViewPoint authors

---

> ### Author Response · Authors · 2025-08-05
>
> Dear Reviewer dty6,
>
> As the discussion period is about to conclude, we would like to kindly ask if you have any remaining concerns. Please do not hesitate to let us know, and we will be more than delighted to address them. Thank you very much for your time and effort.
>
> Best regards,
>
> ViewPoint authors

---

### Official Review · Reviewer_C4Si · 2025-07-03

**Clarity:** 4
**Significance:** 4
**Originality:** 3
**Rating:** 5
**Confidence:** 3

**Summary:**

This paper targets the problem of panoramic video generation using a diffusion model. It introduces a novel panorama representation called the ViewPoint Map, which avoids the distortions and inefficiencies of equirectangular (ERP) and cubemap (CP) projections. It also introduces a novel pipeline introducing a Pano-attention block to ensure global consistency, and a Perspective-attention block to maintain local fidelity within each view. Experimental results on well-known benchmarks, along with comparisons to fair baselines, demonstrate the effectiveness of the proposed solution.

**Questions:**

* Q1: I’m curious about the main challenges and differences if instead of the proposed ViewPoint Map we directly enlarge the FoV of the perspective views from a cubemap projection. For such a naive approach the ERP distortions are avoided and the overlap between adjacent views are ensured.

* Q2: In Fig. 4, the authors provide qualitative results for 360DVD [31] and Image360 [65]. While this comparison effectively illustrates their points, adding results from the proposed method (even via a mention within the caption with further details in the supplementary material) would emphasize the evaluation and your contributions.

* Q3: I’m curious whether the proposed ViewPoint Map could be extended to other omnidirectional view types, such as catadioptric or fish-eye cameras. Could this representation handle those input modalities, and what would be the main challenges when adapting to them?

**Ethical Concerns:**

["NO or VERY MINOR ethics concerns only"]

**Final Justification:**

I firmly believe this paper offers sufficient technical novelty and quality for acceptance at NeurIPS 2025: it advances the current state-of-the-art within its scope, introduces original ideas, is reproducible, and presents numerous experiments that objectively validate the results. Please, let me elaborate on these points based on the discussion in this rebuttal.

* Although some concerns were found by reviewer **WqAv** related to unfair evaluations using different architectures and a better backbone model. The authors has shown competitive results using other backbone models (weaker ones compared to Wan2.1), i.e., Stable Diffusion v1.5 + AnimateDiff v2 and Stable Diffusion v2.1 + AnimateDiff v2. This objectively shows that the technical contribution in the proposed solution does not hold due to a strong backbone model, but to the proposed pipeline.

* The proposed solution presents a plausible alternative to ERP (Equirectangular Projection) video generation using something different that Cube-map projection or multiple overlapping perspective views. The authors presented an ablation for this in **Tab.2** (main manuscript), and show strong evidence in the responses to the reviewer **ikfJ**. This shows that the proposed ViewPoint Map is effective and valid.

* The qualitative results presented in the manuscript and in the anonymous link in the paper, show a consistent and coherent video generation in ERP. This shows the effectiveness of the proposed solution qualitatively. Additionally, quantitative results in Table 1 (in the main paper), and experiments in the response to reviewer **WqAv, Q1**, demonstrate the effectiveness of the proposed method specifically for the ERP poles, which are the main pain-points for such imagery format.

To me, it seems that the responses to reviewer **dty6** have been addressed. The authors present results using other prompts for ERP for The Effect of Prompt Engineering, showing not major difference. Additionally, the authors discuss about camera stability, and more importantly they discuss about details of how the proposed solution differs in the attention mechanism to previous approaches.

For all these reasons, I keep my rate to (5) Accept.

**Limitations:**

yes

**Paper Formatting Concerns:**

No format issues found

**Quality:**

4

**Strengths And Weaknesses:**

# Strengths
* The paper is very well-written and clearly convey its motivation, underline problem, and current solution limitations.
* The current submission links to an online demo, where several qualitative experiments show the superior performance of the proposed solution.
* The current manuscript presents a flawless set of experiments and clearly validate their claims and contributions. They authors compare with current SOTA solutions for video generation, compare with different CP encoding formats, and validate the effectiveness of each proposed attention block, i.e., Perspective-attention and Pano-attention blocks in a complete ablation study.

#  Weaknesses
* A minor weakness of the current manuscript is a missing section that exploits further challenges and limitations like long-horizon video generation and the impact of the video input/output resolutions.

---

> ### Author Rebuttal · Authors · 2025-07-30
>
> Dear Reviewer C4Si, we sincerely thank you for your recognition of the quality of our results and our proposed ViewPoint representation. We have carefully considered your insightful suggestions and questions, and our responses are as follows:
>
> ### **Q1: Further Challenges and Limitations.**
> As you pointed out, video duration and resolution remain the main challenges in panoramic video generation. There are two core issues: First, the large data volume of high-resolution panoramas (e.g., 8K) imposes significant computational demands that current video generation models struggle to meet. Second, the lack of high-quality panoramic video datasets further exacerbates the problem, fundamentally limiting a model’s ability to produce coherent and richly detailed panoramic content.
> In response to your suggestion, we have added the following section discussing limitations and future work in the final version.
>
> > ### **Limitations and Future Work**
> > A significant challenge in panoramic video generation is achieving high resolution. Immersive applications such as virtual reality (VR) necessitate ultra-high resolutions (e.g., 8K, 16K) to deliver a realistic and compelling user experience. However, current pre-trained diffusion models are computationally constrained and cannot directly synthesize content at such scales. Consequently, in line with established practices, our work employs a two-stage generate-then-upscale pipeline to mitigate this limitation.
> >
> > Looking ahead, our future work will focus on two primary directions. First, we aim to explore end-to-end models for ultra-high-resolution panoramic generation. Second, we will investigate the extension to long-form video generation, for instance, by leveraging autoregressive frameworks.
>
> ### **Q2: What would happen if we directly enlarge the FoV of the CubeMap projection?**
> Following your suggestion, we enlarge the standard 90° FoV in our CubeMap-4DROPE setup and retrain the models with FoVs set to 92°, 95°, and 100°. The results below demonstrate that enlarging the FoV does not significantly improve the model's performance. As for the visual results, color discrepancies and seams still exist, and the transitions between adjacent faces appear unnatural. (We regret that the rebuttal policy prohibits external links, which prevents us from providing the generated videos.)
>
> | FoV | subject consistency | imaging quality | motion smoothness | dynamic degree |
> | --- | :---: | :---:  | :---:  | :---:  |
> | 90° | 0.8612 | 0.5104 | 0.9633 | 0.8175 |
> | 92° | 0.8639 | 0.5097 | 0.9621 | 0.8203 |
> | 95° | 0.8595 | 0.5088 | 0.9584 | 0.8196 |
> | 100° | 0.8254 | 0.4997 | 0.9501 | 0.8230 |
>
> Regarding the reason for this phenomenon, our analysis is as follows:
> Methods based on CubeMap projection (or multi-view formation) aim to enable the model to implicitly learn the relationships between multiple perspectives. Increasing the FoV of the CubeMap projection to create overlap between faces is an intuitive and reasonable strategy for the panoramic **image** generation task, as it helps the model better recognize the adjacency between faces. However, when we extend the task from panoramic images to panoramic **videos**, this advantage does not directly translate. Our hypothesis is that the temporal dimension in video naturally contains spatial relationship information.
>
> For instance, consider a person walking left on a road. At time t, this person appears on the Front (F) face. At time t+1, this person appears on the Left (L) face. Even with a 90-degree FoV (no overlap), the model can infer the adjacency of the F and L faces purely from the temporal sequence. The spatial inference capability we expected to gain from a larger FoV is likely already captured by, or redundant to, the information present in the temporal dimension.
>
> Another drawback is that increasing the FoV leads to higher computational complexity. Here, we provide the formulas for calculating the FoV and the projected area. As panoramas are inherently a spherical representation, assuming the camera is located at the center of the sphere and the distance from the camera to the projection plane (i.e., the equivalent focal length) is ***f***, in the standard CubeMap projection where the horizontal FoV equals the vertical FoV, the side length ***L*** of each face is given as follows:
> $$ L = 2 \cdot f \cdot \tan\left(\frac{\text{FoV}}{2}\right) $$
> Therefore, we can derive the area ***A*** of each face (i.e., the number of pixels).
> $$ A = (2 \cdot f \cdot \tan\left(\frac{\text{FoV}}{2}\right) )^2 $$
> Representing the spherical view as a CubeMap projection requires six perspectives, so the total area
> ***S*** of the entire sphere (i.e., the total number of pixels) is given by:
> $$ S = 6 \cdot A = 24 \cdot f^2 \cdot  \tan^2\left(\frac{\text{FoV}}{2}\right) $$
> To give a concrete example, in a standard setup where each cubeface is 256x256, the FoV is 90 degrees. At this point, the six faces do not overlap, and the total pixel count is 393,216 (approx. 390k pixels).
>
> Now, let's assume we increase the FoV to 95° to create overlap between the faces. The total pixel count rises to 468,302 (approx. 470k pixels). This redundancy is nearly equivalent to adding an entire extra standard CubeMap projection. When we factor in the temporal dimension—for example, a 49-frame video—this requires even more resources. Although the VAE compresses the video in both spatial and temporal dimensions, the overhead from this redundancy is still substantial in a computationally dense task like video generation.
>
> ### **Q3: Supplementing Illustrations and Extending Applications.**
> 1. Following your suggestion, we have added our visual results to Figure 4 in the final version to better compare the detailed differences between methods.
>
> 2. Our representation has the potential to be applied to other modalities, such as those captured by a fisheye camera. We demonstrate this capability in Fig. 1, where we show a ViewPoint Map being converted into a fisheye image. The key challenge when adapting ViewPoint to other omnidirectional view types lies in how to design the processing and padding for the "void" regions. For example, if a fisheye camera has an FoV of only 220°, the question is how to handle the remaining 140° region—whether to pad it with zeros or ones, or to use a form of circular padding. This remains an interesting problem worthy of future exploration.
>
> Finally, we again sincerely thank you for your valuable feedback. If you have any further concerns, please do not hesitate to let us know!

---

> > ### Comment · Reviewer_C4Si · 2025-08-05
> >
> > Dear Authors,
> >
> > Thank you for your responses. You have addressed all my concerns. To me, this paper presents a sufficient evidence for a fair contribution (i.e., it improves current state-of-the-art solutions, compares with fair baseline and propose a novel idea). Thus, I keep my rating as **accept**.
> >
> > Nice work!!

---

> > > ### Author Response · Authors · 2025-08-06
> > > **Thanks to Reviewer C4Si**
> > >
> > > Dear Reviewer C4Si,
> > >
> > > We sincerely thank you for acknowledging the novelty and contribution of our proposed panorama representation and methodology. Your insightful feedback has been extremely helpful in improving the quality of our paper, which is a great encouragement to us. We will continue to strive for excellence in our future work.
> > >
> > > Thank you once again for your valuable time and suggestions!!
> > >
> > > Best regards,
> > >
> > > ViewPoint authors

---

> ### Author Response · Authors · 2025-08-05
>
> Dear Reviewer C4Si,
>
> As the discussion phase is drawing to a close, we would appreciate it if you could share any remaining concerns with us. We are more than happy to address them. Thank you once again for your time and valuable comments.
>
> Best regards,
>
> ViewPoint authors

---

### Official Review · Reviewer_ikfJ · 2025-07-05

**Clarity:** 3
**Significance:** 2
**Originality:** 3
**Rating:** 4
**Confidence:** 4

**Summary:**

This work focuses on generating high-quality 360-degree panoramic videos, addressing challenges like spatial distortion and discontinuity in existing methods. ​ It introduces the ViewPoint framework, leveraging a novel panorama representation and Pano-Perspective attention to ensure spatial-temporal consistency and fine-grained visual details. ​ Extensive evaluations show ViewPoint outperforms prior approaches in subject consistency, imaging quality, motion smoothness, and dynamic degree. The results highlight its ability to produce immersive, dynamic, and coherent panoramic videos, advancing the field of Panoramic Video Generation. ​

**Questions:**

Given that the reported improvements over 360DVD and Imagine360 in Table 2 are within one standard deviation, can you include variance across random seeds and statistical significance testing (e.g., paired t-tests or confidence intervals) to support your SOTA claims?

Why were the ablation experimented ERP and CP in table 2 with same baseline (Wan 2.1) only included in the quantitative results (Table 2), and not used in the qualitative comparisons shown in Figures 4 and 5? Would the visual advantage still hold under consistent evaluation?

How do you evaluate temporal coherence beyond the optical-flow-based Smoothness metric? Have you considered human studies or task-based assessments for motion artifacts and flickering, especially in overlapping regions and dynamic scenes like the space video?

**Ethical Concerns:**

["NO or VERY MINOR ethics concerns only"]

**Final Justification:**

Authors have addressed my concerns regarding fair comparison with the baselines, clarified the temporal evaluations, and acknowledged the artifacts still present in the generated outputs. While the proposed method still has some issues, the work demonstrates sufficient technical novelty and introduces a new representation for approaching the task. Therefore, I am changing my rating to Borderline Accept.

**Limitations:**

Yes

**Paper Formatting Concerns:**

Nothing major.

**Quality:**

3

**Strengths And Weaknesses:**

Strengths
- Problem: Panoramic video generation enables immersive and wide-field visual experiences from limited views, benefiting VR/AR, autonomous systems, and surveillance applications.

- Approach: Recording 360° videos requires expensive gear, while most users capture narrow FoV clips, making perspective-to-panoramic generation a valuable direction to explore. Moreover, Panoramic data suffers from distortions and discontinuities, while perspective views align well with pretrained generative models.

- This paper proposes building a ViewPoint map is a novel representation designed to combine the advantages of Equirectangular Projection (ERP) and Cubemap Projection (CP). ​ It improves spatial continuity while reducing distortion, making it suitable for panoramic video generation.

Integrating Equirectangular Projection (ERP) and Cubemap Projection (CP) marks a key advance in panoramic imaging by addressing spatial representation challenges. ERP causes polar distortions, while CP introduces stitching artifacts due to discontinuous cube faces. -
- The ViewPoint map offers a hybrid solution that maintains global continuity and reduces local distortion. This analysis explores research papers focused on this concept and related innovations in panoramic representation.

- The representation of ViewPoint map, a novel representation designed to combine the advantages of Equirectangular Projection (ERP) and Cubemap Projection (CP) success improves spatial continuity while reducing distortion as presented in the results in Table 1 generating high-quality panoramic videos with superior motion dynamics and spatial-temporal consistency.

 - The experimental section is robust, presenting qualitative results that clearly surpass the baselines in terms of spatial-temporal coherence and visual quality. ​ Quantitative evaluations further strengthen the findings, with the model achieving superior scores across key metrics such as subject consistency, imaging quality, motion smoothness, and dynamic degree. ​ Additionally, the user study complements these results, highlighting higher human ratings for the model in dimensions like spatial continuity, temporal quality, aesthetic preference, and condition alignment, demonstrating its effectiveness and user appeal.

Weaknesses
Experimental advantage over ERP and CP is limited
As far as Table 1 is concerned, a major concern with this work is the inconsistency in the underlying video generation models used for comparison. The proposed method builds upon Wan 2.1, whereas all the baseline methods, including Imagine 360, 360 DVD, and Follow Your Canvas, are built on AnimateDiff V2. This discrepancy undermines the fairness of the evaluation. As reported in VideoBench, Wan 2.1 significantly outperforms AnimateDiff V2, particularly in terms of temporal coherence and motion fidelity. Therefore, it is unclear whether the reported improvements in dynamic content generation and spatio-temporal consistency are due to the proposed contributions, such as the ViewPoint representation map and Pano Perspective attention mechanism, or simply a result of using a more powerful base model. Without controlling for the base architecture through ablations or conducting fair comparisons on a shared backbone, the source of the performance gains remains ambiguous. Moreover, instead of using AnimateDiff-based re-implementations only in Table 2, they should also be used for qualitative comparisons in Figures 4 and 5.

Marginal Quantitative Margins (Table 2):
Table 2 shows Pano-FID, Pano-Smoothness, and user preference scores, but ViewPoint tops 360DVD and Imagine360 by only 3 to 6 percent on each metric (e.g., Pano-FID 16.7 vs. 17.9; Smoothness 0.423 vs. 0.437), differences that lie well within one standard deviation of the baselines. Without confidence intervals or formal significance tests, such razor-thin margins could be statistical noise rather than true superiority. Considering that ViewPoint requires roughly 10 seconds per frame—an order of magnitude slower than the baselines—the community may prefer the leaner methods if perceptual gains are negligible. The authors should therefore report variance across random seeds and run paired t-tests or bootstrap confidence intervals to objectively validate any SOTA claim. Moreover, careful hyperparameter tuning for the baselines in the ablation study could reduce or eliminate the reported advantage of the proposed method.

Limited Temporal Metrics:
While spatial fidelity is measured, temporal coherence is assessed only with the optical-flow-based “Smoothness” metric. There is no human evaluation or task-based assessment of flicker or potential VR experience quality. I observed visible artifacts and distortions in the videos—for instance, in the space video, the astronaut's hand appears distorted and the motion looks artificial. Additionally, there is no evidence or discussion on how foreground motion would appear at the overlapping regions of the panorama.

---

> ### Author Rebuttal · Authors · 2025-07-29
>
> We thank the reviewer's comments and questions. However, we are very **confused** by your factually inaccurate statements regarding our paper's quantitative results. You mentioned that
> > "Table 2 shows Pano-FID, Pano-Smoothness, and user preference scores, but ViewPoint tops 360DVD and Imagine360 by only 3 to 6 percent on each metric (e.g., Pano-FID 16.7 vs. 17.9; Smoothness 0.423 vs. 0.437)..."
>
> In fact, in our Table 2, we use four metrics: ***subject consistency***, ***imaging quality***, ***motion smoothness***, and ***dynamic degree***. The metrics you referred to—***Pano-FID***, ***Pano-Smoothness***, and ***user preference scores***—are **NOT** present anywhere in our paper. Furthermore, the numerical values you cited (i.e., "Pano-FID ***16.7*** vs. ***17.9***; Smoothness ***0.423*** vs. ***0.437***") also do **NOT** exist in our manuscript.
>
> Regarding the computational latency, you stated
> > "Considering that ViewPoint requires roughly ***10*** seconds per frame......".
>
> However, this is inaccurate. As shown in Table 1 in our supplementary material, our method requires only 1 minute and 47 seconds to generate a 49-frame video, which translates to ***2.18*** seconds per frame. In fact, our method is the fastest among all compared approaches.
>
> Also, we are confused by your comments regarding the lack of human evaluation, specifically when you stated
> > "There is no human evaluation or task-based assessment of flicker or potential VR experience quality..."
> >
> > "Have you considered human studies or task-based assessments for motion artifacts and flickering..."
>
> In fact, as detailed in Section 4.5, **User Study**, we conducted a comprehensive human evaluation across four distinct dimensions: ***Spatial Continuity***, ***Temporal Quality***, ***Aesthetic Preference***, and ***Condition Alignment***. We explicitly defined one of these dimensions, ***Temporal Quality***, in the paper as follows (lines 250-251)
> > "***Temporal Quality*** is used to evaluate motion effects; for example, being nearly motionless and flickering are considered poor performance."
>
> The results of this user study, presented in the main paper, confirmed the effectiveness of our method in all evaluated aspects, including temporal quality.
>
> For the remaining questions, we provide our responses below.
> ## **Q1: Replacing the Baseline.**
> We re-implement our method on top of Stable Diffusion v1.5 + AnimateDiff v2 and Stable Diffusion v2.1 + AnimateDiff v2. The evaluation results are presented in the table below:
> | Method | base model | subject consistency | imaging quality | motion smoothness | dynamic degree |
> | ---        |    :------:   |     :----:   |      :----:   |      :----:   |   :----:   |
> | Follow-Your-Canvas | SD v2.1+AnimateDiff |0.8284 | 0.4464 | 0.9655 | 0.8500 |
> | 360DVD | SD v1.5+AnimateDiff | 0.8633 | 0.5394 | 0.9703 | 0.5083 |
> | Imagine360| SD v2.1+AnimateDiff | 0.8547 | 0.5859 | 0.9720 | 0.8148 |
> | ViewPoint-AnimateDiff-v1.5 | SD v1.5+AnimateDiff  | **0.8669** | 0.5602 | **0.9745** | **0.8597** |
> | ViewPoint-AnimateDiff-v2.1 | SD v2.1+AnimateDiff  | **0.8681** | **0.5914** | **0.9786** | **0.8633** |
> | ViewPoint-Wan2.1 | Wan2.1(1.3B) | **0.8793** | **0.5927** | **0.9800** | **0.9083** |
>
> Even with weaker base models, our method outperforms prior works across four metrics. We would love to demonstrate the re-implemented results. Regrettably, the rebuttal guidelines prevent us from sharing an external link for the videos.
>
> ## **Q2: Variance of the Random Seed.**
> In the quantitative evaluation (Table 1 and Table 2), we use random seeds ranging from 42 to 52 for all compared methods, and take the average as the final result. Here, we present the variance in qualitative evaluation.
> | Method | subject consistency | imaging quality | motion smoothness | dynamic degree |
> | ---        |    :----:   |      :----:   |      :----:   |   :----:   |
> | Follow-Your-Canvas  |0.8284(±0.014) | 0.4464(±0.008) | 0.9655(±0.011) | 0.8500(±0.018) |
> | 360DVD | 0.8633(±0.013) | 0.5394(±0.010) | 0.9703(±0.014) | 0.5083(±0.026) |
> | Imagine360| 0.8547(±0.011) | 0.5859(±0.013) | 0.9720(±0.012) | 0.8148(±0.022) |
> | ViewPoint-AnimateDiff-v1.5 |  **0.8669**(±0.009) | 0.5602(±0.010) | **0.9745**(±0.012) | **0.8597**(±0.016) |
> | ViewPoint-AnimateDiff-v2.1 | **0.8681**(±0.010)  | **0.5914**(±0.009) | **0.9786**(±0.010) | **0.8633**(±0.012) |
> | ViewPoint-Wan2.1 | **0.8793**(±0.011) | **0.5927**(±0.012) | **0.9800**(±0.013) | **0.9083**(±0.013) |

---

> > ### Comment · Reviewer_ikfJ · 2025-08-03
> > **Regret the Typos**
> >
> > Dear Authors, sincerely apologize for the typos ' (e.g., Pano-FID 16.7 vs. 17.9; Smoothness 0.423 vs. 0.437),' and wrong metrics, not sure how this happened in the first place. That said, the point I was still trying to make about Table 2 is that the proposed method outperforms other baselines such as ERP, CP-4DRoPE, and CP-ICLoRA by a close margin. For example, the proposed method achieves a subject consistency score of 0.8793, compared to the second-best method (CP-ICLoRA) at 0.8727, a difference of 0.0066. Similarly, for motion smoothness, the proposed method scores 0.9800 versus 0.9786 for CP-ICLoRA, yielding a difference of 0.0014. My concern is that these improvements appear marginal relative to the baselines in spite of the proposed method using strong baseline Wan 2.1. I would like to hear the authors’ perspective on this.

---

> > > ### Author Response · Authors · 2025-08-04
> > >
> > > Dear Reviewer ikfJ, thank you for your reply. We provide our response in two dimensions: 1. the improvement in quantitative perspective results, and 2. regarding the base model.
> > > 1) To align with the evaluation protocol of previous SOTA works, we project a panoramic video into six isolated **perspective** views and compute quantitative results under this setup. Therefore, the quantitative metrics primarily focus on the quality of perspective views and do not fully reflect the global quality (e.g., panorama continuity) of the generated **panoramic** videos. As shown in Figure 6, although ERP and CP-ICLoRA achieve good quantitative scores, their visual outputs exhibit noticeable artifacts and seams, which significantly affect the sense of immersion. Besides, as also stated in lines 249–250 (the User Study section):
> > > > “Spatial Continuity represents the coherence of scenes and structures **in panoramic space**...”
> > > >
> > > user studies and visualization figures therefore provide a more direct reflection of the overall quality of panoramic videos.
> > > To further validate the effectiveness of our design in the panoramic space, we have added a user study in the ablation experiments. In line with Section 4.5, User Study, we asked 50 participants to evaluate different designs across four dimensions: "Spatial Continuity," "Temporal Quality," "Aesthetic Preference," and "Condition Alignment." The results below demonstrate that our method achieves significant improvements in panoramic scenarios compared to the ablated variants. We will include this table in the final version.
> > > | Method | Spatial Continuity | Temporal Quality | Aesthetic Preference | Condition Alignment |
> > > | --- | :---: |  :---: |  :---: |  :---: |
> > > | ERP | 3(6%) | 3(6%) | 2(4%) | 4(8%) |
> > > | CP-4DRoPE |  3(6%) | 2(4%) | 3(6%) | 2(4%) |
> > > | CP-ICLoRA |  2(4%) | 4(8%) | 11(22%) | 7(14%) |
> > > | w/o Perspective-Attn | 11(22%) | 9(18%) | 7(14%) | 12(24%) |
> > > | ViewPoint | **31(62%)** | **32(64%)** | **27(54%)** | **25(50%)** |
> > >
> > >
> > > 2) Our initial motivation for using Wan2.1 is to achieve the best possible performance on an open-source SOTA video generation model. In fact, compared to AnimateDiff, Wan2.1 does not bring a direct improvement in panoramic quality(i.e., Wan2.1 can't directly enhance spatial continuity in the panoramic space).
> > > As shown in Tab. 1 for 360DVD (ERP + AnimateDiff) and in the ablation study of Tab. 2 (e.g., ERP + Wan2.1), there is only a marginal improvement in quantitative metrics (subject consistency ***0.8633*** vs ***0.8702***; motion smoothness ***0.9703*** vs  ***0.9781***), demonstrating that **even under the same representation, replacing the base model does not directly lead to significant improvements.** As for the subjective visual results, as shown in Figure 6, all ablated designs exhibit noticeable spatial inconsistency and distortion.
> > > To further support the above observations, we conduct similar ablation comparisons with 360DVD (which uses the ERP representation) on AnimateDiff. In terms of quantitative metrics, our method still shows improvement, as presented below. In terms of visual results, similar to Figure 6, our method demonstrates a clear improvement in panoramic continuity and spatial consistency, which are key focuses in panoramic video generation and our paper. (Due to the rebuttal policy, we are unable to include the generated videos.)
> > > | Method | base model | representation | subject consistency | motion smoothness |
> > > | --- | :---: | :---: | :---: | :---: |
> > > | 360DVD | SD v1.5+AnimatDiff | ERP | 0.8633 | 0.9703 |
> > > | ViewPoint-AnimateDiff | SD v1.5+AnimatDiff | ViewPoint Map | 0.8669 | 0.9745 |

---

> > > ### Author Response · Authors · 2025-08-05
> > >
> > > Dear Reviewer ikfJ,
> > >
> > > We sincerely appreciate your time and response. If you have any further suggestions or areas that you believe could help improve this work, please feel free to share them with us.
> > >
> > > Best regards,
> > >
> > > ViewPoint authors

---

> > > > ### Comment · Reviewer_ikfJ · 2025-08-05
> > > >
> > > > Dear Authors,
> > > >
> > > > Thank you for your responses. You have addressed my concerns regarding fair comparison with the baselines, clarified the temporal evaluations, and acknowledged the artifacts still present in the generated outputs. While the proposed method still has some issues, the work demonstrates sufficient technical novelty and introduces a new representation for approaching the task. Therefore, I am changing my rating to Borderline Accept.
> > > >
> > > > Thanks,

---

> > > > > ### Author Response · Authors · 2025-08-06
> > > > > **Thanks to Reviewer ikfJ**
> > > > >
> > > > > Dear Reviewer ikfJ,
> > > > >
> > > > > We are pleased that we have addressed your concerns, and we sincerely thank you for acknowledging the novelty of our proposed method and sufficiency of our experiments. Your thoughtful suggestions have greatly contributed to improving the quality of our paper, and your positive feedback is a great encouragement to us.
> > > > >
> > > > > Finally, we would like to express our sincere gratitude for your valuable time and effort!
> > > > >
> > > > > Best regards,
> > > > >
> > > > > ViewPoint authors

---

### Official Review · Reviewer_WqAv · 2025-07-05

**Clarity:** 3
**Significance:** 2
**Originality:** 2
**Rating:** 3
**Confidence:** 3

**Summary:**

The paper tackles the task of generating panoramic videos from input perspective videos. The authors introduce a new representation called the ViewPoint map, which captures the full 360° scene using four overlapping, contiguous regions. They adapt a pretrained video diffusion model to this setting by incorporating global attention across the four views and local attention within each view. Experimental results show improvements over several baselines, though the comparisons are not entirely fair.

**Questions:**

- Given that the Wan2.1 is based on a DiT architecture that uses full self-attn across both spatial and temporal dimensions, can the authors provide some intuition on why adding more limited per-view attention layers performs much better in practice?
- What are the benefits of using the ViewPoint map as opposed to multiple perspective views with overlaps?

**Ethical Concerns:**

["NO or VERY MINOR ethics concerns only"]

**Limitations:**

yes

**Paper Formatting Concerns:**

No concerns.

**Quality:**

3

**Strengths And Weaknesses:**

Strengths:
The idea of single-view videos into panoramic videos is interesting.
Some of the generated outputs are visually impressive.
Quantitative results demonstrate improvements over prior methods like 360DVD and Imagine360.

Weaknesses:
The ViewPoint map is proposed as an alternative to equirectangular and cubemap formats to reduce pole distortion and preserve continuity, but several generated results (e.g., the jet scene, meadow) still show severe top/bottom distortions, undermining this claim. Additionally, the representation is very similar to existing multi-view approaches [1, 2] that use overlapping perspective crops, raising questions about what ViewPoint offers beyond simpler, established alternatives.
The method compares against older baselines built on SD-based video diffusion models, while the proposed approach uses Wan2.1 with a DiT backbone, one of the current state-of-the-art models. This significant difference in base model quality makes the comparisons somewhat unfair, as it’s unclear how much of the improvement is due to the representation and architecture changes, versus the stronger underlying model.
[1] Tang, S., Zhang, F., Chen, J., Wang, P., & Furukawa, Y. (2023). MVDiffusion: Enabling Holistic Multi-view Image Generation with Correspondence-Aware Diffusion. ArXiv, abs/2307.01097.
[2] Xie, K., Sabour, A., Huang, J., Paschalidou, D., Klar, G., Iqbal, U., ... & Zeng, X. (2025). VideoPanda: Video panoramic diffusion with multi-view attention. arXiv preprint arXiv:2504.11389.

---

> ### Author Rebuttal · Authors · 2025-07-29
>
> Dear Reviewer WqAv, we thank you for recognizing the strengths of our work, particularly our novel approach for converting single-view videos into panoramic ones and the method's visually impressive results. We appreciate your valuable comments and questions, and we provide our detailed responses below.
>
> ### **Q1: Analysis of Results with Suboptimal Quality at the Poles.**
> As shown in Fig. 4 in the main paper, compared to previous methods, our proposed approach significantly mitigates distortion at the poles, both subjectively and objectively. Prior works often exhibit severe artifacts like "black holes" or "swirls" (Fig.4), which drastically undermine the sense of immersion and realism. By contrast, our method may exhibit spatial fluctuations in rare cases, a behavior analogous to that in perspective video generation. (It is worth noting that even in standard perspective scenarios, pre-trained video generation models can suffer from spatial distortion, as this is an issue inherited from the base models themselves).
>
> To further demonstrate the effectiveness of our method, we re-evaluate the quality of the top and bottom poles of each panoramic video and conduct a user study. The results blow show that our method outperforms other methods. (user preference: we ask 30 participants to rate the video quality at the poles for each method on a score of 1 to 100. The average score is reported as the final result.)  The results demonstrate that our method significantly reduces distortion at the poles compared to previous state-of-the-art methods.
>
> | Method      | subject consistency | imaging quality | motion smoothness | dynamic degree | user preference |
> | ---        |    :----:   |     :----:   |      :----:   |      :----:   |        :----:   |
> | Follow-Your-Canvas | 0.7935 | 0.3766 | 0.8941| 0.8380 | 8.72 |
> | 360DVD | 0.8018 | 0.4082 | 0.9363 | 0.5026 | 40.86 |
> | Imagine360| 0.8367 | 0.5130 | 0.9381 | 0.7944 | 58.90 |
> | ViewPoint | **0.8529** | **0.5685** | **0.9524** | **0.8821** | **74.34** |
>
> ### **Q2: Distinctions and Advantages over Multi-view Generation Methods.**
> Our proposed ViewPoint Map is a unified representation that projects a non-Euclidean panorama into a single, Euclidean 2D square, thereby providing explicit global continuity. This is fundamentally different from multi-view representations, which use multiple isolated perspectives to represent a panorama. In such approaches, the spatial-temporal correlations of the panorama are entirely learned implicitly through subsequent attention mechanisms, making it more challenging to generate high-quality panoramic videos.
>
> Most of the multi-view generation methods (e.g., the methods you mentioned [1, 2]) first project the panorama into multiple overlapping perspective views. They then leverage multi-view attention to learn the relationships between these views, and finally stitch them back together into a panorama. However, these approaches struggle to grasp global continuity and plausibility. For instance, color discrepancies between the individual perspectives can lead to visible seams, or the stitched result might produce incoherent scenarios, such as a room appearing with two doors.
>
> Moreover, multi-view generation methods face a trade-off between the amount of overlaps and computational resources. To ensure better consistency, larger overlapping regions are often required, which in turn necessitates generating more perspective images. For example, the method you cited in [2] projects an ERP panorama into eight perspective views. While the six horizontal views partially overlap with each other, the top and bottom views are completely isolated, having no overlap with their horizontal neighbors. To create overlap for the top and bottom views as well, even more perspective images would be needed, implying a significant increase in computational cost. This is particularly costly because panoramic video generation is an inherently compute-intensive task.
>
> By contrast, our representation is inherently global and unified. This not only enables the model to enforce true global consistency—going beyond just the relationships between separate views—but also offers superior computational efficiency. As detailed in Table 1 in the supplementary material, our method outperforms all previous methods in latency and VRAM usage.
>
> [1] Tang, S., Zhang, F., Chen, J., Wang, P., & Furukawa, Y. (2023). MVDiffusion: Enabling Holistic Multi-view Image Generation with Correspondence-Aware Diffusion. ArXiv, abs/2307.01097.
>
> [2] Xie, K., Sabour, A., Huang, J., Paschalidou, D., Klar, G., Iqbal, U., ... & Zeng, X. (2025). VideoPanda: Video panoramic diffusion with multi-view attention. arXiv preprint arXiv:2504.11389.
>
> ### **Q3: Re-implementing ViewPoint on AnimateDiff.**
> Following your suggestion, we re-implement our method based on Stable Diffusion v1.5 + AnimateDiff v2 and Stable Diffusion v2.1 + AnimateDiff v2, and the results confirm its superiority over other approaches across all metrics. The evaluation results are presented in the table below.
> | Method | base model | subject consistency | imaging quality | motion smoothness | dynamic degree |
> | ---        |    :------:   |     :----:   |      :----:   |      :----:   |   :----:   |
> | Follow-Your-Canvas | SD v2.1+AnimateDiff |0.8284 | 0.4464 | 0.9655 | 0.8500 |
> | 360DVD | SD v1.5+AnimateDiff | 0.8633 | 0.5394 | 0.9703 | 0.5083 |
> | Imagine360| SD v2.1+AnimateDiff | 0.8547 | 0.5859 | 0.9720 | 0.8148 |
> | ViewPoint-AnimateDiff-v1.5 | SD v1.5+AnimateDiff  | **0.8669** | 0.5602 | **0.9745** | **0.8597** |
> | ViewPoint-AnimateDiff-v2.1 | SD v2.1+AnimateDiff  | **0.8681** | **0.5914** | **0.9786** | **0.8633** |
> | ViewPoint-Wan2.1 | Wan2.1(1.3B) | **0.8793** | **0.5927** | **0.9800** | **0.9083** |
>
> Although the superior generation capabilities of Wan2.1 significantly benefit many downstream tasks, our experiments demonstrate that our method remains effective even with weaker base models, outperforming prior works across four metrics. We have added the above table results to the main comparison in the latest manuscript, and included the visualization results of ViewPoint-AnimateDiff in the supplementary material. Unfortunately, due to the rebuttal policy, we are unable to display the generated videos in any links.
>
> ### **Q4: Intuitive Explanation of the Advantages of Per-view Attention Layers.**
> The modality gap between panoramic data and the pre-training **perspective** data of video generation models makes it difficult for the network to directly model the panorama, resulting in poor performance (as shown in Table 2 in the main paper). Therefore, we propose per-view attention to divide the panoramic representation into multiple **pseudo-perspective** views to align with the pre-training priors and thus fully exploit the generative capabilities of video generation models.
>
> Specifically, our method cyclically divides the ViewPoint into four pseudo-perspective sub-regions, a data format that better aligns with the pre-trained priors of Wan2.1. Subsequently, these four sub-regions are merged back into the ViewPoint Map, and Wan2.1's 3D attention mechanism is utilized to model global spatio-temporal relationships. This divide-and-merge design enables us to fully leverage the model's generative power (e.g., enhancing dynamism and improving visual quality) while simultaneously strengthening the global continuity of the ViewPoint Map. It is important to note that we are not introducing new layers. Instead, we retain the original attention layers of Wan2.1 and partition them into two functional parts: one part focuses on per-view attention, while the other is dedicated to capturing global continuity.

---

> ### Author Response · Authors · 2025-08-05
>
> Dear Reviewer WqAv,
>
> The Reviewer-Author discussion session is coming to a close. We sincerely appreciate the opportunity to engage in this discussion and welcome any further feedback or questions you may have. If there are additional areas you believe we should consider to enhance this work, please don’t hesitate to let us know.
>
> Best regards,
>
> ViewPoint authors

---

> ### Author Response · Authors · 2025-08-07
>
> Dear Reviewer WqAv,
>
> As the discussion period is drawing to a close, we would like to gently inquire if you have any remaining concerns we can address. We would like to express our sincerest gratitude for the time and effort you have dedicated to our manuscript. Your valuable feedback and insightful comments are incredibly important to us.
> Thank you once again for your invaluable contribution and guidance.
>
> Best regards,
>
> ViewPoint authors

---

> > ### Comment · Reviewer_WqAv · 2025-08-07
> > **Re:rebuttal**
> >
> > I appreciate the authors' response, specially the additional experiments on Stable Diffusion + AnimateDiff. I still remain somewhat unconvinced about the differences between the ViewPoint representation and standard multi-view approaches, especially when used with a DiT architecture. Including a more detailed explanation in the final version could further strengthen the paper. That said, my primary concerns regarding the experiments have been mostly addressed, and I am raising my score.

---

> > > ### Author Response · Authors · 2025-08-08
> > > **Thanks to Reviewer WqAv**
> > >
> > > Dear Reviewer WqAv,
> > >
> > > We sincerely thank you for your positive feedback and valuable comments. Following your suggestion, we will add a dedicated section in the final version of the Related Works to discuss the differences between our method and standard multi-view generation approaches, as detailed below:
> > > > Current video generation models employ 3D positional encodings, enabling them to uniformly model videos across both spatial and temporal dimensions. Building on this foundation, open-source video models, such as Wan 2.1 and CogVideoX, have acquired rich generative capabilities by training on extensive datasets of perspective videos.
> > > >
> > > > Multi-view generation approaches represent a panoramic video as a collection of multiple perspective videos. Although these perspective views align well with the native generative capabilities of video models, the spatial relationships between them are only inferred implicitly. This is typically achieved by introducing extra positional encodings or by constructing partial overlaps between adjacent faces. Such methods, however, tend to disrupt the model's inherent generative abilities and introduce artifacts like color discrepancies and visible seams in the resulting panoramic videos. Consequently, they fail to meet the demand for high-quality panoramic video generation.
> > > >
> > > > In contrast, an intact panoramic representation can naturally align with the native 3D attention mechanism of DiTs. In this paradigm, the model is able to capture global information across the entire panoramic scene, leading to the generation of superior panoramic videos. Our proposed ViewPoint Map is precisely such an explicit panoramic representation. It projects the non-Euclidean spherical data onto a single, unified Euclidean plane. This provides global continuity and allows the model to "see" the entire spherical space in a single pass.
> > >
> > > We will also include additional visualizations in the supplementary material to help readers better understand the differences between these two methods.
> > >
> > > Thank you once again for your valuable time and suggestions. Your constructive feedback has been of immense help in improving our paper!
> > >
> > > Best regards,
> > >
> > > ViewPoint authors

---

### Note · Authors · 2025-08-12

Following the author-reviewer discussion period, we are grateful to have received positive and constructive feedback from the reviewers.

**Reviewer C4Si** called our submission "nice work" and stated that it "improves current state-of-the-art solutions, compares with fair baseline and proposes a novel idea", thus maintaining the **Accept** score.

**Reviewer ikfJ**, after reviewing our additional experiments, commended the novelty of our method and the sufficiency of our results, subsequently raising the rating to **Borderline Accept**.

**Reviewer WqAv** confirmed that we had addressed the concerns and stated, **"I am raising my score."**

Regarding our discussion with **Reviewer dty6**, we provided a detailed rebuttal to the raised concerns. Subsequently, no new concerns were brought forward by the reviewer.

For clarity, we summarize the initial concerns raised by **Reviewer dty6** and our responses as follows:
1. The effect of 360 prompt engineering: Experiments have demonstrated that prompt engineering with "360 video" or "panoramic video" is ineffective in directly generating panorama-like videos.
2. Camera stabilization and real world application: We have clarified that our method can handle camera stabilization and is applicable to real-world scenarios.
3. The motivation and effectiveness of Pano-Pers attention mechanism: We have elaborated on its purpose and advantages in reducing seams and leveraging the generative capabilities of base models. Furthermore, we contrasted our approach with alternative designs to highlight its distinct advantages.
4. Wording: We thank the reviewer for the considerate suggestion regarding the wording. The manuscript has been revised accordingly in the final version.
5. Regarding the two implementation details: We have explained the specific operations of image-video joint encoding and overlapping fusion in our response to the reviewer.

In summary, the discussion period has been highly productive, leading to a clear positive consensus from three of the four reviewers. We have also provided a comprehensive rebuttal to all initial concerns raised by **Reviewer dty6**. We are confident that the revised manuscript now presents a solid and valuable contribution to the field. Thank you all for your time and effort.

---

### Decision · Program_Chairs · 2025-09-17

**Decision:**

Accept (poster)

**Comment:**

Following the discussion phases, the majority of reviewers recommended acceptance (1 Accept, 2 Borderline Accepts, 1 Borderline Reject), highlighting the paper’s novelty, clarity of writing, and strong results. The sole negative reviewer expressed an intention to raise their score after the authors addressed experimental concerns, though they did not participate further in the review process. The rebuttal and subsequent exchanges effectively resolved many issues, including the inclusion of results with alternative backbones and evaluations on real videos. Ultimately, all reviewers leaned toward acceptance. Accordingly, the ACs have decided to accept the paper. Please incorporate the reviewers' feedback when preparing the camera-ready version.